# MC-SSL: Towards Multi-Concept Self-Supervised Learning

## Abstract

Self-supervised pretraining is the method of choice for natural language processing models and is rapidly gaining popularity in many vision tasks. Recently, self-supervised pretraining has shown to outperform supervised pretraining for many downstream vision applications, marking a milestone in the area. This superiority is attributed to the negative impact of incomplete labelling of the training images, which convey multiple concepts, but are annotated using a single dominant class label. Although Self-Supervised Learning (SSL), in principle, is free of this limitation, the choice of a pretext task facilitating SSL can perpetuate this shortcoming by driving the learning process towards a single concept output. This study aims to investigate the possibility of modelling all the concepts present in an image without using labels. In this respect, the proposed Multi-Concept SSL (MC-SSL) framework is a step towards unsupervised learning, which embraces all the diverse content in an image with the aim of explicitly modelling the information from all the concepts present in the image.

MC-SSL involves two core design steps: Group Mask Model Learning (GMML) and learning of pseudo-concepts for data tokens using a momentum encoder framework. The experimental results on multi-label and multi-class image classification downstream tasks demonstrate that MC-SSL not only surpasses existing SSL methods but also outperforms supervised transfer learning. The source code will be made publicly available for community to train on bigger corpus.

## 1 Introduction

Recent advances in self-supervised learning [1, 2, 3, 4, 5, 6] have shown great promise for downstream applications, particularly for image classification datasets with labels highlighting one dominant concept per image (also known as multi-class datasets, *e.g.* ImageNet-1K [7]). A typical representative of these SSL approaches, i.e. DINO [6], trains the system to extract and associate image features with a single dominant concept in a promising way, but it ignores the intrinsic multi-label nature of natural images that depict more than one object/concept. The empirical evidence provided by Stock and Cisse [8] study proved that the remaining error in the ImageNet-1K dataset is predominantly due to the single-label annotation. Indeed every pixel in an image is part of some semantic concept. However, collecting an exhaustive list of labels for every object/concept is not scalable and requires significant human effort, making large scale supervised training infeasible for multi-label datasets. In view of this, it is pertinent to ask what the best strategy is to address these deficiencies. We believe that multi-concept self-supervised learning (i.e. ability to represent each concept/object in an image without using labels) is the principled way forward.

This study is a step towards building a self-supervised (SS) framework that is capable of learning representations for all the objects in an image. Once the multiple-concepts are extracted by MC-SSL with no labels, the expensive multi-label annotated datasets will only be needed for evaluation.

MC-SSL is built on Group Mask Model Learning (GMML) introduced in [1, 9]. In MC-SSL, the network is trained with two objectives: i) reconstruct the raw pixel values of the GMML-based manipulated data-tokens and, crucially, ii) learn patch level concepts/classes for individual data-tokens. MC-SSL forces the network to learn the representation of an object/concept, i.e. its properties such as colour, texture and structure, as well as context, in order to reconstruct and recover the distorted data-tokens by using the information conveyed by unmasked data tokens. This encourages all the

Figure 1: MC-SSL has basic knowledge of objects as shown by the self-learnt grouping of data-tokens by k-means clustering by only providing number of clusters the output data-token from vision transformer should be clustered to. Data tokens on objects of the same class have similar representation. This is achieved as a by-product of MC-SSL without any labels or any self-supervised segmentation objective. Notice how the concepts are refined when asking for more concepts to be discovered.To achieve concept learning is still a challenge for MC-SSL as shown by the spread out representation exemplified by the $5^{th}$ column for example. This demands training on bigger datasets and the design of even more advanced MC-SSL methods, which this research aims to stimulate.

data-tokens on an object to have similar representation and to incorporate local context (refer to Figure 1). The ultimate role of the auxiliary but complementary task of learning a patch classifier is to assign a pseudo-semantic label to a group of context aware data tokens covering an object. Our conjecture is that learning pseudo labels for patches encourages data tokens on similar objects within an image and across images to belong to the same pseudo class, promoting intra and inter image concept consistency. Our scientific hypothesis is that this context based learning of objects across a collection of images through representation clustering will conform to human semantic labels.

The main contribution of the work is the introduction of a novel SSL framework which models the information conveyed by all the objects/concepts present in the image rather than focusing on the dominant object. Most importantly, from the practical point of view, MC-SSL enables the training of data hungry transformers from scratch with only a few thousand images. The possibility of training from scratch on limited data with high accuracy will have a significant impact on small AI research groups, companies, and application domains which are relying on the use of pretrained models. Additionally, we show that, although MC-SSL is unaware of the semantic concepts present in an image, the self-learnt grouping of data-tokens corresponds to distinct semantic concepts as evident from Figure 1. The impact of the proposed innovation is that MC-SSL outperforms state-of-the-art (SOTA) SSL methods by a large margin in multi-label classification tasks, and achieves competitive results in multi-class tasks. Lastly, MC-SSL based self-supervised pretraining outperforms supervised pretraining for downstream tasks.

## 2 RELATED WORK

Self-supervised methods can roughly be categorised to generative and discriminative approaches. Generative approaches [10, 11, 12] learn to model the distribution of the data. However, data distribution modelling generally is computationally expensive and may not be necessary for representation learning in all scenarios. On the other hand, discriminative approaches, typically implemented in a contrastive learning framework [13, 14, 15, 16, 3, 2, 6, 17, 18, 19] or using pre-text tasks [20, 21, 22, 1, 5, 23], demonstrate the ability to obtain better generalised representations with modest computational requirements.

A large body of work on contrastive learning trains the model to discriminate between images considering each of them as a different class. Such approaches require either large batches [15, 19] or memory banks [13, 24] to perform well which is often computationally expensive.

Another recent line of work [2, 25, 3, 17, 18, 6] has shown the feasibility to learn feature representations that are invariant to different augmented views of the same image, without requiring negative samples. Such approaches are prone to learn trivial embeddings. Grill *et al.* [3] prevent a mode collapse by employing a momentum encoder with an asymmetric predictor. Barlow Twins [17] and VICREG [18] employ a covariance constraint. In particular, in Barlow Twins, the model is trained to obtain an identity cross-correlation matrix between the outputs of two identical networks fed with

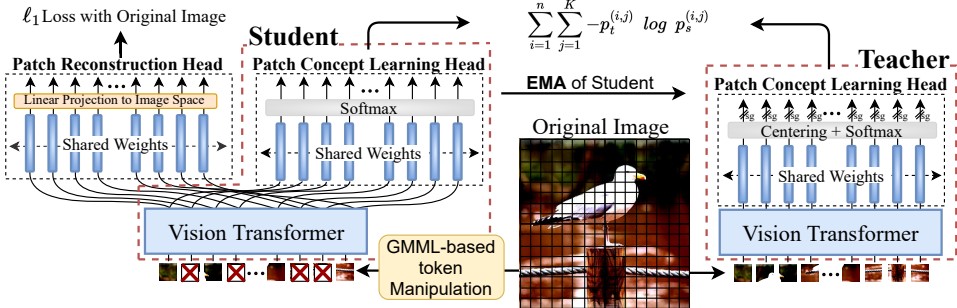

Figure 2: Proposed MC-SSL framework; A step towards Self-supervised Transformers for Multi-Concept Learning. MC-SSL consists of two principles, (a) recovery of transformed information which is realised by patch reconstruction head ($s_{pr}(.)$ left block/head with $\ell_1$ loss), and (b) learning of pseudo-concepts by patch concept learning module ($s_{pc}(.)$ block/head of the student network). The recovery of the transformed information from the non-transformed data-tokens by $s_{pr}(.)$ indicates that the network has learnt the semantics of different concepts in terms of contextual understanding and learnt useful inductive bias by learning local statistical correlation in natural images. While the role of $s_{pc}(.)$ is to map the data-tokens from the same concepts to a common pseudo-concept space ensuring representation consistency among different instances of the same concept within an image and across images.

the augmented versions of a given image. In contrast, Caron *et al.* [6] proposed centring trick by preventing one dimension to dominate.

Despite the impressive results achieved by contrastive learning methods, some of them tend to encourage modelling of one dominant class per image [26] using holistic representation and/or disregard the learning of contextual representations, for which alternative pretext tasks, such as inpainting patches [27], colourisation [28, 29, 20], relative patch location [10], solving jigsaw puzzles [30, 31], predicting image rotations [21], spotting artefacts [22] etc., might be better suited.

One of the pioneering work combining self-supervised learning and vision transformers is SiT [1]. SiT introduced two key concepts for self-supervised learning of vision transformers 1) Group Mask Model Learning (GMML) and 2) Use of masked auto-encoder. The idea of GMML is to learn a neural network by transforming a group of connected patches and recover them by using visible contextual data. Several works adopted the idea of GMML, e.g., in vision [32, 23, 5], multimodal analysis [33], medical imaging [34], audio [35], anomaly detection [36]. Beit [5] employed the GMML but instead of recovering the pixel values, the network is trained to predict the class of visual tokens corresponding to the masked patches using the publicly available image tokenizer described in [37]. Beit can be considered as an extreme case of data-token level distillation via a network pretrained in a self-supervisory way. Nevertheless, Xie *et al.* [32] and He *et al.* [23] demonstrated that the use of cumbersome pretrained tokenizer for mapping data patches to class token is not needed, and even the simple reconstruction with GMML outperforms Beit. Different from existing SSL works, we propose MC-SSL, a novel SSL framework based on the idea of GMML for masked autoencoder and learning of pseudo-concepts for data-tokens based on knowledge distillation.

## 3 METHODOLOGY

In this section, we introduce a self-supervised framework which is a step towards multi-concept self-supervised learning (MC-SSL). Our proposed framework is based on GMML and self-learning of data token conceptualisation with the incorporation of knowledge distillation [38]. In knowledge distillation a student network $s^\theta(.)$ is trained to match the output of a given teacher network $t^\phi(.)$, where $\theta$ and $\phi$ are the parameters of the student and the teacher networks, respectively. In this work, we employ the same network architecture for both the student and the teacher (i.e. the teacher is a momentum encoder [24] of the student), where the parameters of the teacher network are updated from the past iterations of the student network using exponential moving average of the student weights with the following update rule: $\phi = \lambda\phi + (1 - \lambda)\theta$.

The network architecture of the student network comprises three components: A vision transformer backbone $s_b(.)$, followed by two projection heads attached to the output of the transformer. One projection head is for patch reconstruction $s_{pr}(.)$, and the second projection head is for patch concept learning $s_{pc}(.)$. The main architecture of our proposed approach is shown in Figure 2.

## 3.1 SELF-SUPERVISED VISION TRANSFORMER

Similar to [39], we use vision transformer, which receives as input a feature map from the output of a convolutional block/layer. The convolutional block takes an input image $\mathbf{x} \in \mathbb{R}^{H \times W \times C}$ and converts it to feature maps of size $\mathbf{x^f} \in \mathbb{R}^{\sqrt{n} \times \sqrt{n} \times D}$, where, $H$, $W$, and $C$ are height, width and channels of the input image, $n$ is the total number of spatial locations in the feature maps and $D$ is the number of feature map channels. Each spatial location in the input feature maps is considered as an input data token to the transformer, yielding a total of $n$ tokens. Both the input and output of the transformer have the same number of dimensions, i.e. $\mathbf{x^f}, \mathbf{y} \in \mathbb{R}^{n \times D}$.

## 3.2 SELF-SUPERVISED TASKS

We adopted GMML as the key component of MC-SSL. In NLP, a single data-token can represent a semantic concept, hence, an effective pretext task is to randomly mask the data-tokens and learn to recover them. However, this naive masking might not be particularly effective in computer vision (CV) as a single data token may not represent a semantic concept. The idea of GMML is to transform a group of connected patches representing a significant part of an object having semantic meaning and recover them by learning a model. To consolidate the semantically related data-tokens forming an object, GMML is performed by applying several transformations to locally connected data-token/patches of the image, including random drop (i.e. replacing random connected patches with noise) and random replace (i.e. replacing random connected patches with patches from another image). Note that unlike in NLP, another distinction of GMML in CV is that a partial transformation of a data-token is possible. The transformer is trained to recover the values of the corrupted/missing data-token to learn better semantic representations of the input images.

It should be noted that these transformed tokens can either be on the dominant (foreground) object or on the non-dominant (background) objects, and recovering these tokens are equally important. Our hypothesis is that there is no such thing as background in natural images. Each patch in the image represents some concept with visual semantic meaning. The intuition is that by modelling all semantic concepts, the network will generalise better for unseen tasks, whether they are related to an object, a distributed object, or to the entire visual signal.

We leverage the strength of the transformer and train it using GMML with two different objectives: 1) Patch reconstruction where the network is trained to reconstruct the image corrupted by the GMML-based manipulation, and 2) Patch concept classification by training the network to learn data-token level concepts/classes for individual data-tokens.

**Patch Reconstruction:** For image reconstruction, we propose to use the transformer as a group masked autoencoder, *i.e.*, visual transformer autoencoder with GMML. By analogy to auto-encoders, our network is trained to reconstruct the input image through the output tokens of the transformer. The GMML-based manipulated images $\bar{\mathbf{x}} \in \mathbb{R}^{H \times W \times C}$ are fed to the student backbone and the output tokens are fed to the patch reconstruction projection head $s_{pr}(.)$ to obtain $\mathbf{x_r} := s_{pr}(s_b(\bar{\mathbf{x}}))$, i.e. the reconstructed image. The $s_{pr}(.)$ projection head consists of 3 fully connected layers; first two with 2048 neurons and GeLU [40] non-linearity each, and the last bottleneck layer with 256 neurons, followed by a transposed convolution to return back to the image space.

The objective of the image reconstruction is to restore the original image from the corrupted image. For this task, we use the $\ell 1$-loss between the original and the reconstructed image $\mathcal{L}_{\mathrm{recons}}(\theta) = ||\mathbf{x} - \mathbf{x_r}||$, where $||.||$ is the $\ell 1$ norm. Although, $\ell 2$-loss generally converges faster than $\ell 1$-loss, $\ell 2$-loss is prone to over-smooth the edges in the restored image [41]. Therefore, $\ell 1$-loss is commonly used for image-to-image processing rather than the $\ell 2$-loss. We compute the loss only on the corrupted pixels, similar to [42, 9].

**Patch Concept Learning:** The idea of concept learning using SSL is inspired by DINO [6]. DINO provides a pseudo label for the student network by setting a low temperature in the softmax activation of the teacher network. This low temperature sharpens the probability vector leading to one class

getting significantly higher probability than the others. Hence, DINO is focusing on learning the dominant concept in the image. Following our hypothesis that modelling only the dominant class in an image can lead to sub-optimal representation learning, we investigate the role of learning the appropriate concept for each of the data token. This is a step towards extracting a representation corresponding to each concept in an image. In a recent work, Beit [5] assigned fixed classes to each of the patches using a pretrained image tokenizer [37]. Then they used BERT/SiT like framework to predict the classes corresponding to a masked data token. Unlike Beit, we employ a momentum encoder (teacher) to generate pseudo labels for the visual tokens, and force the student model to make predictions consistent with the teacher model. The patch concept learning gives the flexibility to adapt to the visual concepts present in images rather than using a pretrained fixed tokenizer. This is an import ingredient in learning the semantic representation of each object, which will be discussed at the end of the section.

In order to generate pseudo labels for the visual tokens, the training images are fed to the backbone of the teacher network, and the outputs of the data tokens are then fed to a patch concept learning projection head to obtain $z_t := t_{pc}(t_b(\mathbf{x})) \in \mathbb{R}^{n \times K}$, where $K$ represents the number of classes of visual tokens. Similar to DINO, the patch concept learning projection head consists of 3 fully connected layers; first two with 2048 neurons and GeLU non-linearity each, and the last bottleneck layer with 256 neurons. The output of the bottleneck layer is $l2$ normalised and directly connected to a weight-normalised fully connected classification layer with $K$ neurons. For each training sample, the GMML-based manipulated images are passed to the student network to obtain $z_s = s_{pc}(s_b(\bar{\mathbf{x}})) \in \mathbb{R}^{n \times K}$. The task is to match $pc_s$ to $pc_t$ employing the Kullback-Leibler divergence (KL) between the outputs of the teacher and student networks.

Training the student to match the teacher output can easily lead to a trivial constant (i.e. collapsed) embeddings. To avoid the model collapse, we adopted the centring and sharpening of the momentum teacher outputs introduced in [6]. Centring encourages the output to follow a uniform distribution, while the sharpening has the opposite effect. Applying both operations balances their effects, which is sufficient to avoid a collapse in the presence of a momentum teacher. The centre $c$ is updated using an exponential moving average over the teacher output. The sharpening is obtained by using a low value for the temperature $\tau_t$ in the teacher softmax normalisation. The output probability distributions $p_t$ and $p_s$ from the teacher and the student networks over $n$ patches and $K$ dimensions are obtained as follows:

$$p_s^{(i,j)} = \frac{\exp(z_s^{(i,j)}/\tau_s)}{\sum_{k=1}^{K} \exp(z_s^{(i,k)}/\tau_s)} \qquad (1) \qquad p_t^{(i,j)} = \frac{\exp(z_t^{(i,j)}/\tau_t)}{\sum_{k=1}^{K} \exp(z_t^{(i,k)}/\tau_t)} \qquad (2)$$

where $z_s$ and $z_t$ are the class logits for the student and the teacher, $p_s(i, .)$ and $p_t(i, .)$ are the probabilities corresponding to the $i^{\text{th}}$ token output by the student and teacher, and $\tau_t$ and $\tau_s$ are the temperature parameters for the teacher and the student, respectively.

$$\mathcal{L}_{\text{classify}}(\theta) = \frac{1}{n \times K} \sum_{i=1}^{n} \sum_{j=1}^{K} -p_t^{(i,j)} \log p_s^{(i,j)} \qquad (3)$$

### 3.3 Putting together the MC-SSL framework

The reconstruction of the grouped masked tokens and learning of patch level concepts with MC-SSL gives us a mechanism to adaptively learn multiple concepts in each image. The hypothesis of MC-SSL is that a network will be able to recover randomly selected but semantically transformed groups of data-tokens from the rest of the semantic clues about the object and context only if it learns the "semantics" of the objects in the image. Once the transformer autoencoder is aware of a semantic concept, the role of the auxiliary patch concept learning task is to assign a shared/common pseudo label corresponding to all semantically related data-tokens. This consolidated information from all the data-tokens about an object (distributed concept) in an image can be assigned a name which humans are using in their daily life. As a data-token represents a small portion of image/object and is likely to have less overall variability, we do not need to learn a large number of classes/concepts for patches, i.e., $K$ can be small. In contrast, a complex object needs to consolidate information from multiple data-tokens, which constitute the object. Hence, even with a small/limited number of local concepts for each data-token, the possible representation space for objects is huge. For example, if the object consists of only four data-tokens, and a patch concept learning space is only 1024, the

number of possible configurations of local concepts will be $1024^4 > 1.9 trillions$. However, due to the local stationarity of natural visual signals the actual combination will be far less.

Another strength of MC-SSL is its attractive learning behaviour. Unlike, other SSL losses, the MC-SSL performance saturates slower, leading to continuous improvement in performance, presumably because a large portion of image is masked, providing heavy augmentation for the training data. Nevertheless, the performance gain becomes marginal after a few hundred epochs.

The transformer autoencoder and the auxiliary task of patch concept learning[1] in MC-SSL provide a superior way to use the information present in an image. More importantly, they jointly create a mechanism to model all the concept/object present in an image. Some self learnt concepts in each images are shown in Figure 1. Specifically, we obtain $y_t := t_b(\mathbf{x}) \in \mathbb{R}^{n \times D}$ which are the output features corresponding to the data-tokens of the input image. The output features are then clustered, employing simple k-means [44]), into 3 and 4 clusters, where each colour represents a different cluster. Note that the model is trained without any sort of supervision, yet, MC-SSL demonstrates the ability to differentiate between concepts. As shown in Section 4, this way of utilising the information enables us to obtain remarkable results with extremely limited resources.

### 3.4 PROPERTIES OF MC-SSL

Some of the attractive properties of MC-SSL include:

1. Training Transformers on tiny datasets: Transformers trained by supervised losses can attend to all the data-tokens coming from an image empowering them to model global information. However, the side effect is that they need a lot more data to model local context. MC-SSL overcomes this limitation by masking group of semantically related data-tokens. This masking encourages the network to learn from semantic information present in the locality of each group of masked tokens to recover the missing information and, hence, to model local contextual information. Therefore, MC-SSL based pretraining makes it possible to train the transformers on small datasets with high accuracy.
2. The MC-SSL framework is aware of the notion of concepts in the sense that the heavily masked information can be reconstructed with respect to shape and texture from the available information in the visible data tokens relating to the same concept/object and surrounding concepts. This is also evident from the ability of MC-SSL to self-cluster semantically related data-tokens from an object in the image without using any labels for training as demonstrated in Figure 1.
3. MC-SSL based pretraining outperforms supervised pretraining for multiple downstream tasks of multi-label and multi-class datasets, given the same amount of pretraining data. Moreover, MC-SSL pretraining also outperforms state-of-the-art SSL pretraining. This is due to the fact that MC-SSL makes a better use of rich semantic information present in all the data-tokens relating to each concept/class/object in an image.
4. A big batch size is a standard requirement in many of the constrastive learning based SSL, making it unusable when relying on a modest GPU resource. MC-SSL does not suffer from this problem and outperforms the SOTA for small batch sizes as well. This strength is coming from the nature of the pretext task, which does not involve negative classes.
5. The proposed framework is generic and is applicable to other application domains of AI, e.g., sound, medical image analysis, etc. We left this for future study.

**Limitations:** Even though MC-SSL establishes itself as SSL SOTA and outperforms supervised pretraining with a significant margin, it is merely a step towards the bigger goal of unsupervised multi-concept representation learning. Although MC-SSL is agnostic of different concepts, it does not build an explicit representation for each concept in an image.

## 4 EXPERIMENTAL RESULTS

We follow the common evaluation protocol to demonstrate the generalisation of the learnt features of the proposed MC-SSL approach by pretraining the model in an unsupervised fashion, followed by

---

[1]We note that in parallel with us, iBot [43] used the idea of patch concept learning. However, iBot focuses on learning a dominant concept via the classification token with DINO loss and hence, models the dominant class, which we think is a limitation of existing SSL methods.

finetuning on a downstream task. We conduct several experiments on multi-class and multi-label image classification downstream tasks and show the performance in Section 4.1 and 4.2, respectively. Further, we conduct several ablation studies to investigate the effect of the individual components of the proposed approach in Section 4.3. We also provide the details of the employed datasets, evaluation metrics, experimental setups, and implementation details of the proposed MC-SSL method in Appendix A and further insights and visualisations in Appendix B.

## 4.1 MULTI-CLASS CLASSIFICATION

We conduct our experimental analysis on standard multi-class classification problems concerned with the recognition of objects in an unconstrained background. For the multi-class classification task, we include a class projection head similar to DINO [6] in the pretraining stage to capture the general representation of the image. For the finetuning, we rely on the vision transformer developer's default hyper-parameters [45].

Table 1 shows the performance of MC-SSL when the models are pretrained and finetuned on small datasets without using any external/additional data. We employed small variant of vision transformer (ViT-S/16) and pretrained all the models for 3000 epochs, as the datasets are very small, except for MAE where we pretrained the models for 6000 epochs for a fair comparison.

MC-SSL outperforms SOTA SSL with a large margin of $4.1\%$, $8.2\%$, $4.1\%$, $4.4\%$, and $0.6\%$ on Flowers, Pets, CUB, Aircraft, and Cars, respectively. In general, contrastive SSL approaches require special data-specific design or hyper-parameter tuning [46], making them not suitable for small datasets which justify the poor performance of DINO in Table 1. The high performance of MC-SSL on small stand-alone datsts is attributed to modelling of local statistical correlations (which is missing in transformers) and global information dictated by the data itself. Even though MAE is also using masked image modelling, the design choice of MAE models the inductive bias mainly in the heavy decoder which is then passed down to last layers of encoder. This counter intuitive way to modelling information makes MAE not suitable for small datasets.

Table 1: Performance of MC-SSL when pretrained and finetuned on the target dataset, i.e. no external data is used, employing ViT-S/16.

| Method | Flowers | Pets | CUB | Aircraft | Cars |
|---|---|---|---|---|---|
| *Random init.* | 68.8 | 45.5 | 25.3 | 29.7 | 26.3 |
| DINO [6] | 72.3 | 63.1 | 53.8 | 60.7 | 75.3 |
| MAE [23] | 86.9 | 73.0 | 59.4 | 69.0 | 91.0 |
| MCSSL | **91.0** | **81.2** | **63.5** | **73.4** | **91.6** |

Further, in Table 2 and 3, we show the ability of MC-SSL when pretrained on large-scale dataset, i.e. ImageNet-1K, and compare the results with SOTA on several downstream tasks. The results demonstrated the generalisation ability of the learnt features by MC-SSL, achieving SOTA accuracies on all of the downstream tasks, including ImageNet-1K.

Table 2: Comparisons with SOTA on ImageNet-1K.

| Method | top-1 acc. |
|---|---|
| ViT-S/16 | |
| Supervised [45] | 79.9 |
| MoCo-v3 [19] | 81.4 |
| DINO [6] | 81.5 |
| MC-SSL | **82.4** |
| ViT-B/16 | |
| Supervised [45] | 82.3 |
| DINO [6] | 82.8 |
| MoCo-v3 [19] | 83.2 |
| Beit [5] | 83.2 |
| MAE [23] | 83.4 |
| SimMIM [32] | 83.6 |
| MC-SSL | **84.0** |

Table 3: Domain Transfer of MC-SSL pretrained on ImageNet-1K.

| Method | Flowers | Pets | CUB | Aircraft | Cars | STL10 |
|---|---|---|---|---|---|---|
| | | | ViT-S/16 | | | |
| *Random init.* | 68.8 | 45.5 | 25.3 | 29.7 | 26.3 | – |
| Supervised [45] | 98.1 | 91.1 | 82.7 | 80.8 | 91.7 | 98.2 |
| DINO* [6] | 97.8 | 89.4 | 80.8 | 83.8 | 93.1 | 96.7 |
| MCSSL | **98.2** | **93.2** | **83.3** | **85.3** | **93.2** | **98.8** |
| | | | ViT-S/8 | | | |
| DINO* [6] | 98.7 | 92.2 | **86.7** | 89.7 | 94.5 | 98.2 |
| MCSSL | **99.1** | **93.8** | **86.7** | **90.8** | **94.8** | **98.9** |
| | | | ViT-B/16 | | | |
| DINO [6] | 98.4 | 90.2 | 80.7 | 81.5 | 93.0 | – |
| SimMIM* [32] | 97.2 | 92.3 | 81.8 | 83.4 | 91.5 | 97.8 |
| MAE* [23] | **98.9** | 92.8 | 84.2 | **88.4** | **93.5** | 98.2 |
| MCSSL | **98.9** | **93.5** | **85.1** | **88.4** | **93.5** | **99.0** |

## 4.2 Multi-Label Classification

To the best of our knowledge, most of the SOTA SSL methods are validated on multi-class downstream tasks. Only a few SSL methods [13, 47] involve multi-label classification as a downstream task using a simple CNN architecture, but not a transformer.

In Table 4, we report the performance of the proposed MC-SSL on three different datasets, namely PASCAL VOC, VG-500, and MS-COCO. First, we show the results when ViT-S/16 is trained from scratch on the downstream task. Then, we show the performance when the model is pretrained and finetuned without using any external dataset. Finally, we report the performance when the models are pretrained with ImageNet-1K employing MC-SSL, MoCo v3, and DINO frameworks, and then finetuned on the multi-label downstream datasets.

As expected, training from random initialisation has produced low accuracies as the amount of data available is insufficient to train the data-hungry vision transformer. The results significantly improved when the models are pretrained using MC-SSL without any external data with $+32.7$, $+26.6$, and $+7.7$ absolute mAP improvement in PASCAL VOC, MS-COCO, and VG-500 datasets, respectively. Further improvement is obtained when MC-SSL is pre-trained on ImageNet-1K, outperforming SSL SOTA frameworks with $+0.9$, $+1.5$, and $+1.5$ absolute mAP improvement in PASCAL VOC, MS-COCO, and VG-500, respectively.

Table 4: mAP (mean Average Precision) of regular inference on multi-label datasets. * are run by us using official pretrained weights. All the models are pretrained using ViT-S/16 (unless mentioned otherwise) with $224 \times 224$ input resolutions and finetuned with $448 \times 448$ resolution.

| Method | PASCALVOC | MSCOCO | VG500 |
|---|---|---|---|
| | pretraining using the given dataset | | |
| Supervised training from scratch | 34.1 | 47.9 | 23.7 |
| SSL pretraining (MC-SSL) | **66.8** | **74.5** | **31.4** |
| | pretraining using MSCOCO+VG500 dataset | | |
| SSL pretraining (DINO[6])* | 82.7 | 72.0 | 30.4 |
| SSL pretraining (MC-SSL) | **86.6** | **75.3** | **32.0** |
| | pretraining using ImageNet-1K dataset | | |
| Supervised pretraining (ResNet-101 [48]) | **92.9** | 78.6 | 30.9 |
| Supervised pretraining (ViT-S/16 [45])* | 92.6 | 81.4 | 33.0 |
| SSL pretraining (DeepCluster − [AlexNet] [13]) | 73.7 | – | – |
| SSL pretraining (SimCLR − [ResNet-50] [15]) | 84.1 | – | – |
| SSL pretraining (MoCo v3 [19])* | 86.0 | 77.8 | 32.3 |
| SSL pretraining (DINO [6])* | 91.6 | 80.8 | 33.4 |
| SSL pretraining (MC-SSL) | 92.5 | **82.3** | **34.9** |

## 4.3 Ablation Studies

Due to the limited resources, all the ablations are conducted on class-wise randomly sampled 10% of ImageNet-1K for pretraining and evaluated on the full validation set of ImageNet-1K. The models are pretrained for 200 epochs employing the ViT-S/16 architecture as the backbone of the student and the teacher of MC-SSL.

**Effect of percentage of image corruption.** Figure 3-a shows the performance on class-wise randomly sampled 10% of ImageNet-1K when MCSSL is pretrained with different percentages of corruption. We found that the optimal ratio is between 50% to 90%. This behaviour was expected as the masking encourages the network to learn semantic information from the clean patches surrounding the groups of masked tokens in order to recover the missing information.

**Effect of longer pretraining.** Figure 3-b shows the performance when the model is pretrained for longer. We found that longer pretraining improves the performance of MC-SSL, where the accuracy is steadily improving even after 800 epochs of pretraining.

**Effect of the output dimension of the patch classification head.** Figure 3-c, shows that size of the output dimension of the PCL head has minor impact on the performance with 1024 being optimal.

**Effect of different components of MC-SSL.** In this set of experiments, we validate the effect of different components of MC-SSL on several downstream tasks in Table 5. Note that the first two rows are corresponding to DINO [6] where only the DINO head is used and GMML [9] where only the image corruption and reconstruction head are used.

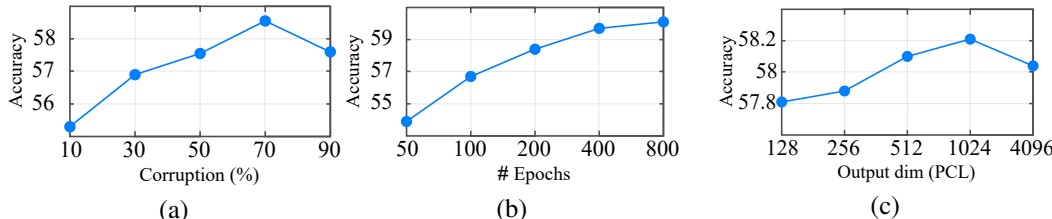

Figure 3: Ablation studies of the effect of (a) Percentage of the corruption, (b) Longer pretraining, and (c) output dimension of the Patch Concept Learning head (PCL).

In summary, both the patch reconstruction and patch concept learning losses with GMML on their own provide the means for SSL, producing an effective starting point for efficient downstream task finetuning. Further improvements are obtained by combining the two in MC-SSL framework. Most importantly, the last two rows of Table 5 show that adding DINO head to the MC-SSL architecture boost the performance on the multi-class classification tasks but degrade the performance on the multi-label tasks.

Table 5: Effect of the different components of MC-SSL for SS pretraining. Models are pretrained on 10% of ImageNet-1K dataset for 200 epochs, followed by finetuning on downstream tasks.

| Method | DINO Head | Image Corrup. | Patch CL | Recons Head | Transfer Learning | | | | | |
|---|---|---|---|---|---|---|---|---|---|---|
| | | | | | Flowers | Pets | STL10 | Cars | INet (10%) | PASCALVOC |
| Dino [6] | ✓ | ✗ | ✗ | ✗ | 95.7 | 84.1 | 95.3 | 88.5 | 55.2 | 80.3 |
| GMML [9] | ✗ | ✓ | ✗ | ✓ | 96.2 | 85.4 | 95.3 | 90.0 | 57.3 | 82.1 |
| | ✗ | ✓ | ✓ | ✗ | 94.4 | 85.9 | 95.5 | 89.3 | 57.0 | 81.7 |
| MCSSL | ✗ | ✓ | ✓ | ✓ | 96.2 | 86.4 | 95.5 | 90.2 | 57.8 | **83.4** |
| | ✓ | ✓ | ✓ | ✓ | **96.6** | **87.0** | **95.8** | **90.6** | **58.1** | 82.5 |

## 5 CONCLUSION AND DISCUSSION

In this paper, we presented a novel self-supervised learning framework (MC-SSL) that enables the extraction of distinct visual representations for multiple objects in an image without annotations. We demonstrated several advantages of the proposed MC-SSL framework. First, MC-SSL can train transformers from scratch with good accuracy on small datasets. Second, MC-SSL has some notion of semantic information as demonstrated by the ability to reconstruct missing parts of a concept and by self-learnt grouping of data-tokens corresponding to a semantic concept (Figure 1). Third, MC-SSL outperforms supervised methods for network pretraining. Last, MC-SSL outperforms the existing state-of-the-art for both multi-class and multi-label downstream tasks, verifying its strengths.

SSL in CV has made a tremendous progress with self-supervised pretraining, outperforming supervised pretraining. However, there are several open questions, which should be addressed in the future development of SSL. We only pose a few of them for brevity. a) Is the kNN style evaluation of SSL methods right? b) What should be the preferred choice to evaluate linear probing and downstream applications? c) Will more suitable evaluation protocols encourage the community to build SSL algorithms for multi-concept representation learning (i.e. algorithms capable to represent each concept/object in an image without using labels)? d) Is it possible to build a representation for each of the concept in an image without any label?

The current kNN and linear evaluation of SSL methods on multi-class datasets, like ImageNet-1K, is biasing the SSL research towards modelling the dominant object in each image, leading to a sub-optimal use of information present in the image. We argue that multi-label datasets like Pascal VOC, Visual Genome, MS COCO and dense prediction datasets are more suitable for evaluating the merits and generalisation capability of SSL methods. The multi-label datasets are suitable for linear probing and for downstream tasks. However, a new kNN evaluation protocol is needed for multi-label classification tasks. Our initial results (Figure 1 and visualisations in appendices) demonstrated the promising direction of building a representation for each of the concepts/objects in an image without any label, and expect that future research will achieve this ultimate goal.

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

## A  EXPERIMENTAL SETTINGS

### A.1  IMPLEMENTATION DETAILS

The backbone architecture of MC-SSL employs a small (ViT-S) and base (ViT-B) variants of vision transformer [39]. Our data augmentation policy follows the SiT [1] policy, which includes random resized cropping, horizontal flipping, followed by randomly applied color jittering, converting to grey-scale, blurring, random drop, and random replace. For the optimisation of the self-supervised training, the model is trained using the Adam optimiser [49] with a batch size of 64 images per GPU,

and momentum of $0.9$. The weight decay follows a cosine schedule [50] from $0.04$ to $0.4$, and the base learning rate is $5e^{-4}$. All the models are trained for $800$. Similar to DINO [6], the sharpening parameters of the teacher and the student are set to $\tau_t = 0.04$ and $\tau_s = 0.1$. The teacher is updated using exponential moving average of the student weights with $\lambda$ following a cosine schedule from $0.996$ to $1$ during training. We combined the patch concept learning and the patch reconstruction losses using simple averaging. We believe further improvements can be gained by optimising the weighted sum of the losses or by incorporating the uncertainty weighting approach [51]. Multi-crop training is an important component of DINO [6] to obtain good features. Although it is not required in our proposed approach, we also adopted the multi-crop strategy to match the number of network updates and have unbiased comparison with DINO. Lastly, for the downstream tasks, the projection heads are discarded and finetuning is performed employing the backbone of the pretrained teacher network $t_b(.)$.

The pyTorch-style pseudocode of MC-SSL is shown in Algorithm 1.

---

**Algorithm 1** MC-SSL PyTorch-style pseudocode.

---

```
#t_b(.),s_b(.): backbone of teacher and student
#t_pr(.),s_pr(.): patch reconstruction projector
#t_pc(.),s_pc(.): patch classification projector
#temp_t,temp_s: teacher & student temperatures
#mc,mt: encoder and center momentum rate
lambda_ = 1
for X in dataloader: # load batch with N samples
    X_clean = augment(X) # N x C x H x W
    X_corrupt = corrupt(X_clean) # N x C x H x W

    # compute projections
    z_t = t_pc(t_b(X_clean)) # N x K
    z_s = s_pc(s_b(X_corrupt)) # N x K
    x_r = s_pr(s_b(X_corrupt)) # N x C x H x W

    #### compute patch classification loss
    z_t_d = z_t.detach() # stop gradient
    # center + sharpen
    z_t_d = softmax((z_t_d - C) / temp_t, dim=1)
    z_s = softmax(z_s / temp_s, dim=1)
    Loss_pc = - (z_t_d * log(z_s)).sum(dim=1).mean()

    #### compute patch reconstruction loss
    Loss_pr = L1Loss(x_r, X_clean)

    # calculate overall loss
    loss =  Loss_pc + lambda_*Loss_pr

    # optimisation step
    loss.backward()
    optimiser.step()

    # update teacher weights
    t.params = mt*t.params + (1 - mt)*s.params

    # update C
    C = mc*C + (1 - mc)*z_t.mean(dim=0)
```

---

## A.2 EXPERIMENTAL SETUP

The models are pre-trained in unsupervised fashion on the ImageNet-1K dataset, with an input size $224 \times 224$. For the finetuning step downstream tasks, the projection heads are replaced with a fully connected layer with $2048$ neurons with the GeLU activation function, followed by an output

layer with $c$ nodes, corresponding to the number of classes in the downstream tasks, followed by a Sigmoid activation function.

For the optimisation of the finetuning, we mostly rely on the vision transformer developer's default hyper-parameters [45] due to the limited resources.

For the multi-class datasets, the models are finetuned with an input size of $224 \times 224$ for 1000 epochs with 64 batch size employing 1 Nvidia Tesla V100 32GB GPU cards.

For the multi-label datasets, we follow the most common settings [52, 53, 54] where the images are resized to $448 \times 448$ and augmented with Rand-Augment [55]. All the models are finetuned for 80 epochs with 48 batch size employing 1 Nvidia Tesla V100 32GB GPU cards. As for the loss function, we adopted the asymmetric loss [56] to address the sample imbalance problem. Asymmetric loss is a variant of focal loss with different $\gamma$ for positive and negative values. Given the target $t = [t_1, t_2, \ldots, t_c] \in \{0, 1\}$, where $c$ is the number of classes, and the output probabilities $p = [p_1, p_2, \ldots, p_c]$, the asymmetric loss for each training sample is calculated as follows:

$$\mathcal{L} = \frac{1}{c} \sum_{i=l}^{c} \begin{cases} (1 - p_i)^{\gamma^+} \log(p_i), & t_i = 1, \\ (p_i)^{\gamma^-} \log(1 - p_i), & t_i = 0 \end{cases} \tag{4}$$

We employed the default values for $\gamma$ in our experiments, *i.e.*, we set $\gamma^+ = 0$ and $\gamma^- = 4$.

### A.3 DATASETS

To evaluate MC-SSL on multi-class downstream tasks, we employed 6 different datasets. The details about the employed datasets are shown in Table 6.

Table 6: Statistics of the employed datasets.

| Dataset | # Classes | #Training Samples | # Testing Samples |
|---------|-----------|-------------------|-------------------|
| Multi-class datasets | | | |
| Flowers [57] | 102 | 2040 | 6149 |
| Pets [58] | 37 | 3680 | 3669 |
| CUB200 [59] | 200 | 5994 | 5794 |
| Aircraft [60] | 100 | 6667 | 3333 |
| Cars [61] | 196 | 8144 | 8041 |
| ImageNet-1K[62] | 1000 | 1.28M | 50,000 |

For multi-label, we conduct our experimental analysis on three datasets, i.e., PASCAL VOC [63], Visual Genome [64], and MS-COCO [65]. The PASCAL VOC 2007 [63] includes 20 object categories and it consists of $5,011$ images for training and $4,952$ for evaluation. The Visual Genome dataset [64] contains $108,077$ images from around 80K categories. We employed VG-500, introduced in [52], consists of $98,249$ training images and $10,000$ test images including the most 500 frequent objects. MS-COCO [65] is a large-scale object detection and segmentation dataset consisting of $82,081$ images for training and $40,137$ images for evaluation. The standard multi-label formulation for MS-COCO includes 80 objects.

# B VISUALISATION

## B.1 COMPARISON WITH THE STATE-OF-THE-ART

To show that MC-SSL has a basic knowledge of objects, we cluster the output features of the self-supervised pre-trained model employing k-means algorithm and compare the results with two strong self-supervised methods, MoCo v3 [15] and DINO [6]. As shown in Fig. 4 and Fig. 5, MC-SSL assigns a similar representation to the data tokens on the same concept without any labels. We used the official pre-trained ViT-S/16 models of MoCo v3 [15] and DINO [6] to obtain the output features of the data-tokens. Besides the ability to learn a superior representation, as evident from the visualisation, an important observation is the continuity and consistency of the learnt representation of data-tokens for the same objects. Unlike MoCo-v3 and DINO, the proposed MC-SSL has less unconnected data-tokens assigned to other objects from a different region of image in the middle of one object. We attribute this consistent and smooth data-token level representation learning to the GMML. Note that we do not employ any explicit consistency learning between data-tokens. Due to the limited resources, MC-SSL is not fully trained and we expect further improvements with more training as evident from the performed ablation in Section 4.

Fig. 6 shows the merits of the MC-SSL for the cases where there are multi instance of the same object category. The challenge here to see whether or not the SSL algorithms can learn the same representation for different instances of the same objects. For a simple case like third column, all methods including MoCo v3 and DINO are working well. However, for complicated cases like the first column and the fourth column, MoCo v3 and DINO are struggling while MC-SSL is still able to maintain decent performance. For example, in the case of the fourth column we notice that there are three semantic concepts: water, ducks and white crane. MC-SSL is able to learn separate representations for water, ducks and crane. Most importantly, the representation for ducks and crane are consistent within their categories. Similarly, in the first column, the representations for deers, land and trees are well learnt by MC-SSL, while other methods struggle.

From the random selection of images and their visualisation, we see the strength of MC-SSL and its significant advantage over existing SSL methods. However, MC-SSL is just a starting point towards learning multi-concepts representations without labels and it is far from human level performance in SSL in the sense that human are accurate and robust in building representations for concepts they have not seen before. In order to show the limitations of MC-SSL, we laboriously went through the generated visualisations and handpicked the examples where MC-SSSL did not perform well as shown in Fig. 7. For example, in Column 1 the green leaves in some cases have similar representation as the trunk of tree, in Column 2, human has a representation similar to desks, in Column 4, machine and door have similar representations, in Column 5 the hat and jacket of the person have similar representation as the walls of the room. In the last column, the fabric on dog is confused with fabric of the car and tongue of the dog is also confused with parts of car.

## B.2 REPRESENTATION ACROSS IMAGES

In the Fig. 8 to Fig. 14, we show that MC-SSL learns representation that are consistent across different images. For simplicity, we cluster the images into two concepts which are self learnt. First self-learnt cluster is attending the dominant object (shown by yellow colour in the centre column) and other is focusing on the environment (shown by blue colour in centre column). The bar plots are showing the probability of the learnt patch concepts corresponding to the dominant object. As can be seen the learnt patch concepts are consist across different instances of semantic concept in different images, enforcing the hypothesis of MC-SSL.

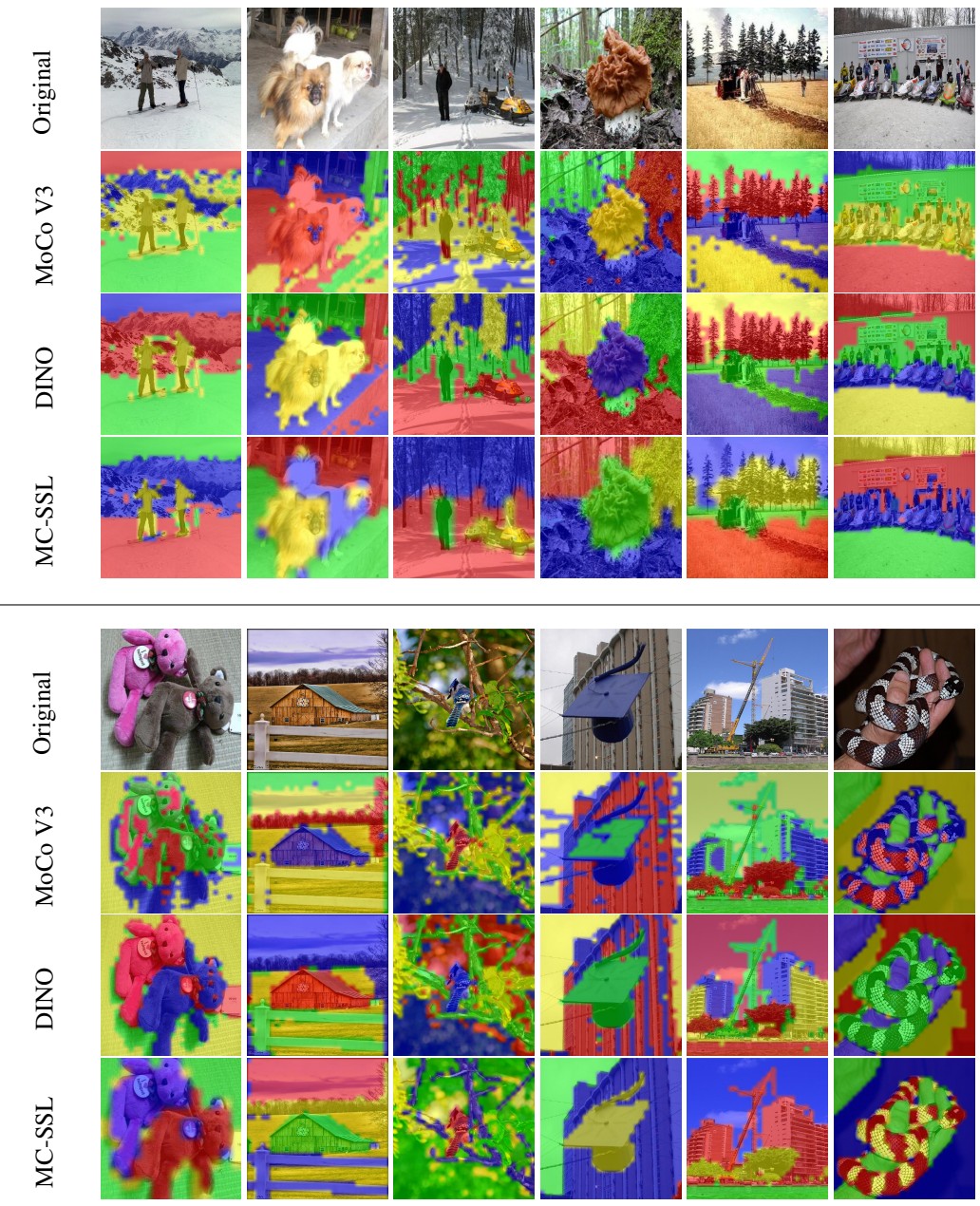

Figure 4: Clustering the output features of the self-supervised pre-trained models into 4 clusters employing k-mean algorithm, where each cluster is represented by a different colour. The original images are shown in the first row. The second, third, and fourth rows show the output clusters of MoCo v3 [15], DINO [6], and MC-SSL (ours), respectively.

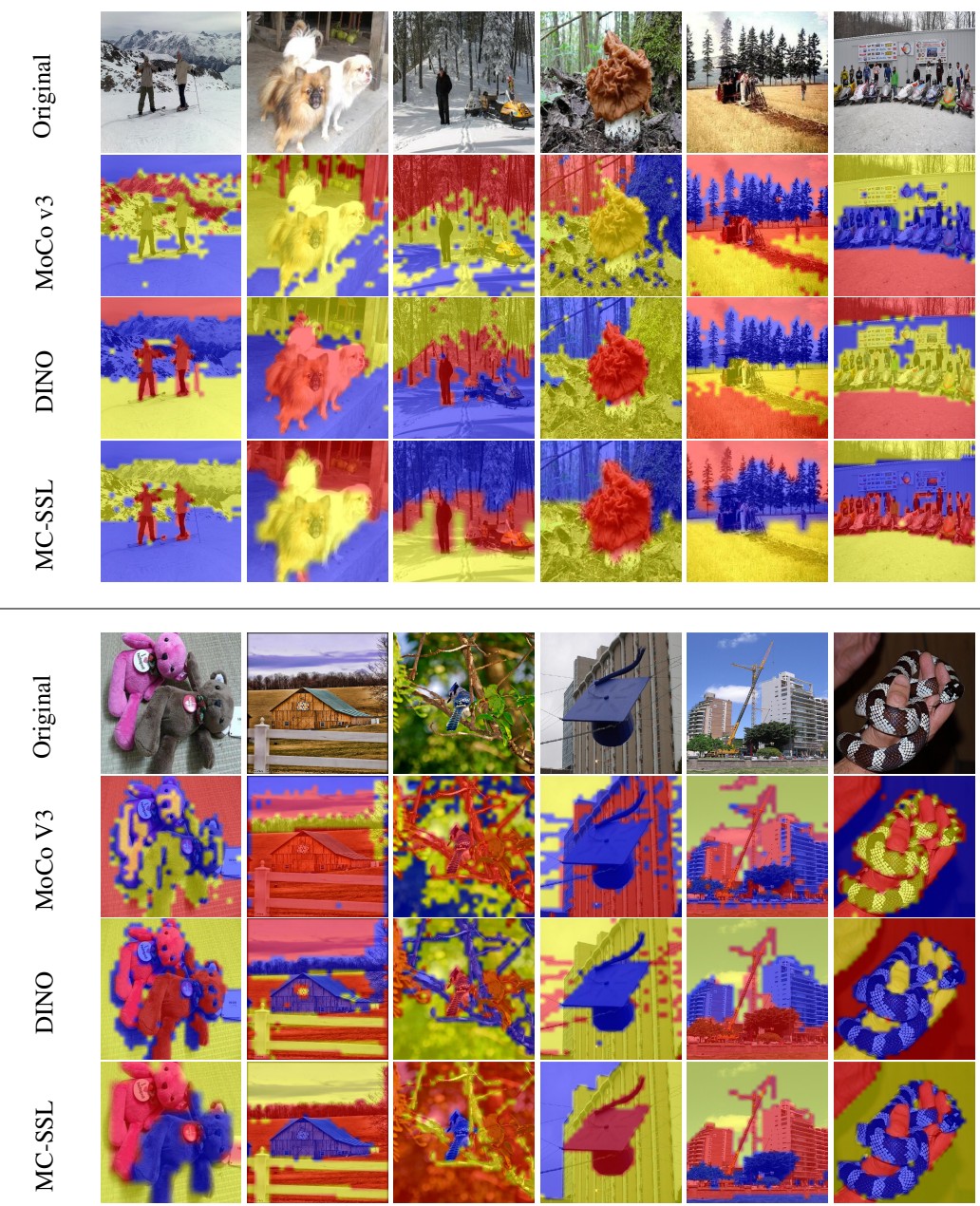

Figure 5: Clustering the output features of the self-supervised pre-trained models into 3 clusters employing k-mean algorithm, where each cluster is represented by a different colour. The original images are shown in the first row. The second, third, and fourth rows show the output clusters of MoCo v3 [15], DINO [6], and MC-SSL (ours), respectively.

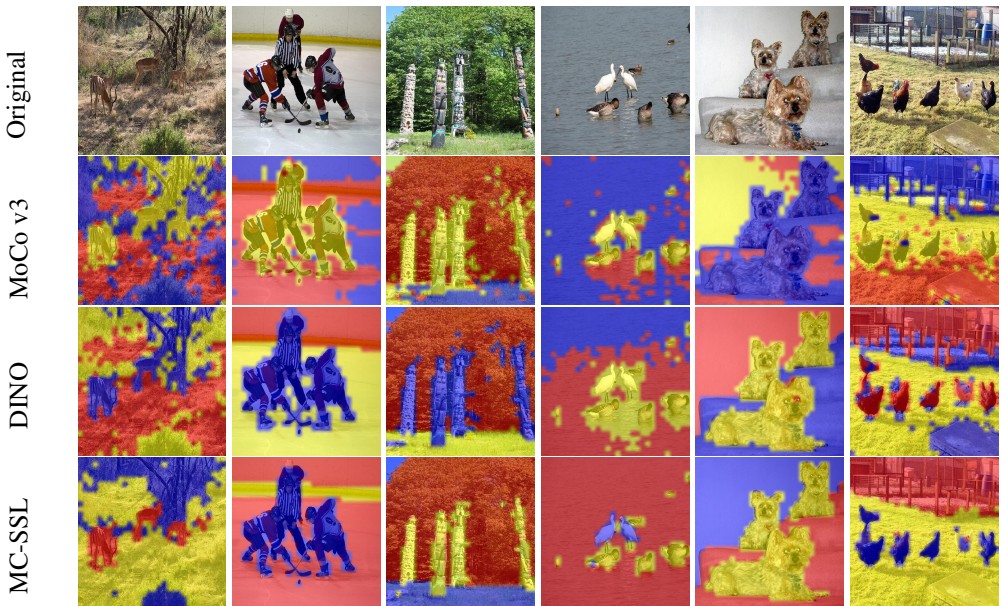

Figure 6: MC-SSL learns consistent representations for data-tokens on the different instances of same object category. Clustering the output features of the self-supervised pre-trained models into 3 clusters employing k-mean algorithm, where each cluster is represented by a different colour.

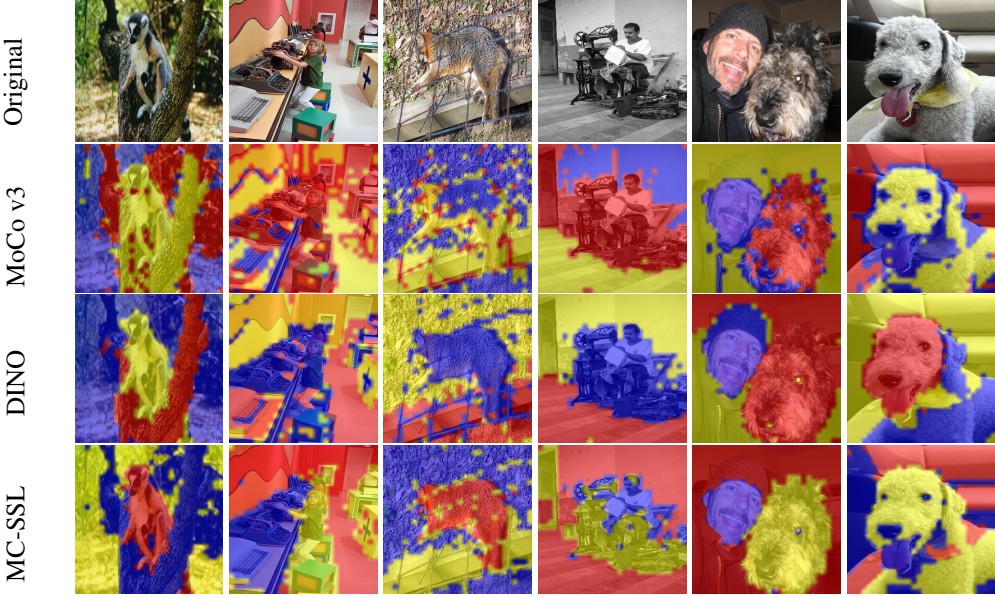

Figure 7: Laboriously handpicked samples to show limitations of MC-SSL. Clustering the output features of the self-supervised pre-trained models into 3 clusters employing k-mean algorithm, where each cluster is represented by a different colour.

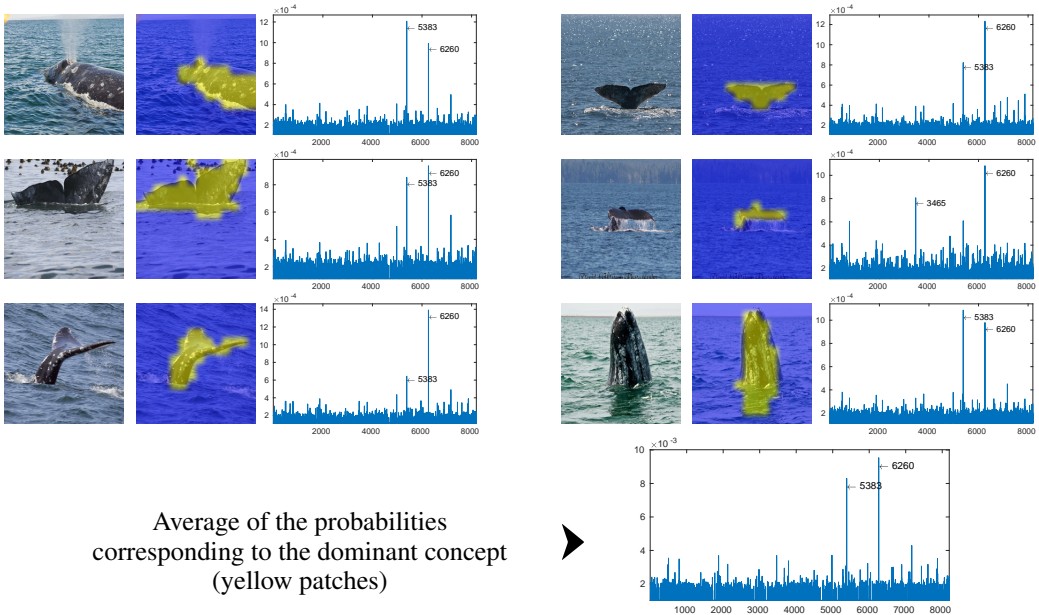

Figure 8: The image is clustered into two concepts which are self learnt. First self-learnt cluster is attending the dominant object (shown by yellow colour in centre column) and other is focusing on the environment (shown by blue colour in centre column). The bar plots are showing the probability of the learnt patch concepts corresponding to the dominant object.

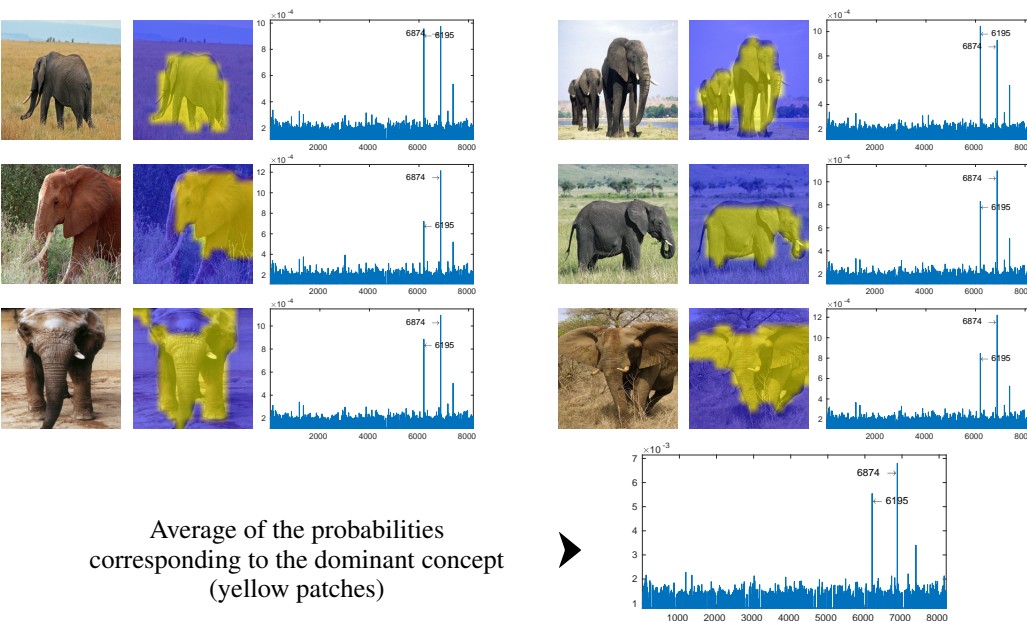

Figure 9: Concept representation across different images.

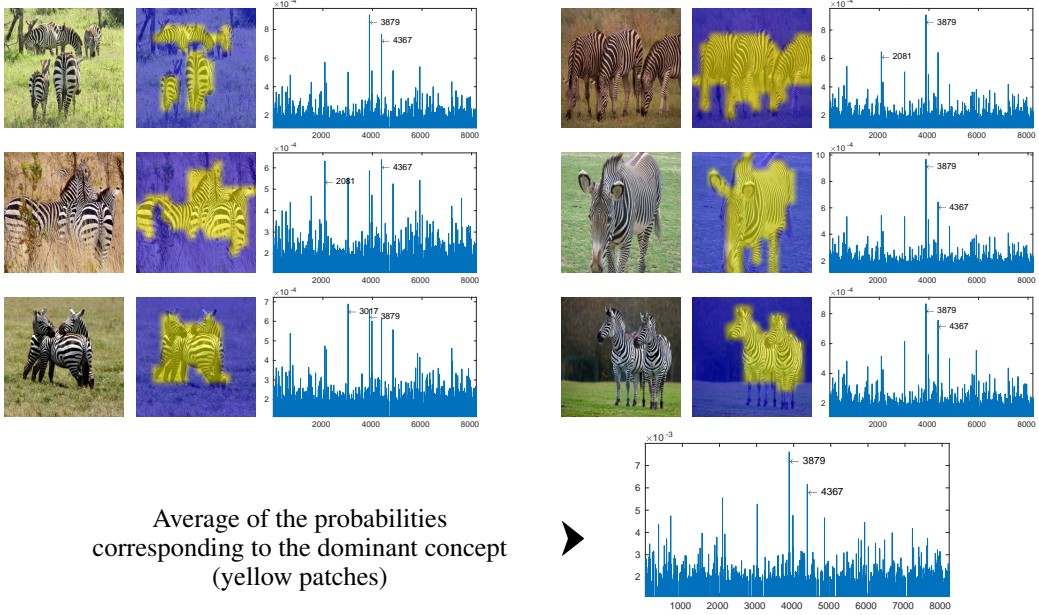

Figure 10: Concept representation across different images.

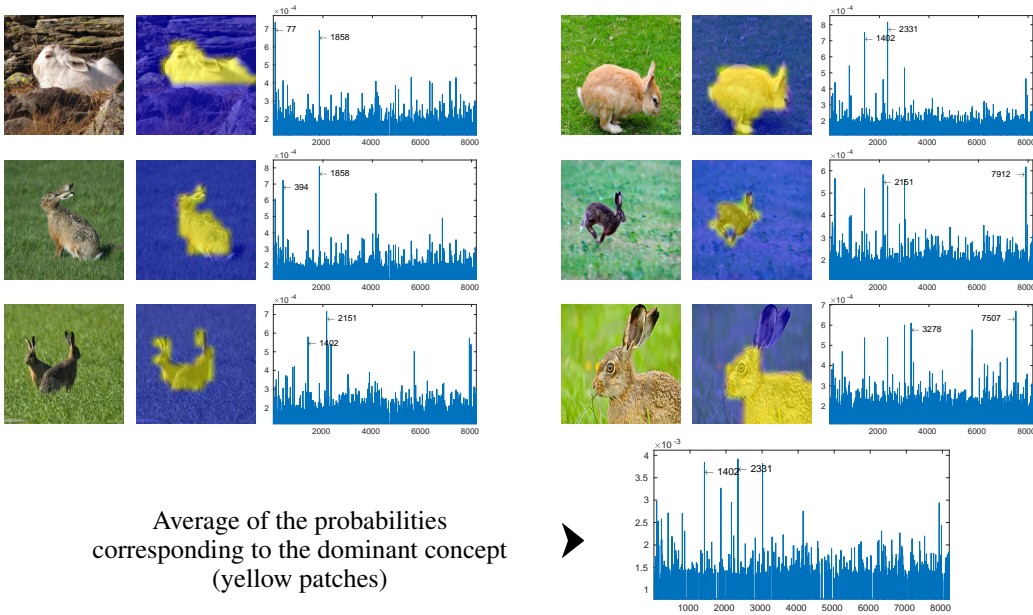

Figure 11: Concept representation across different images.

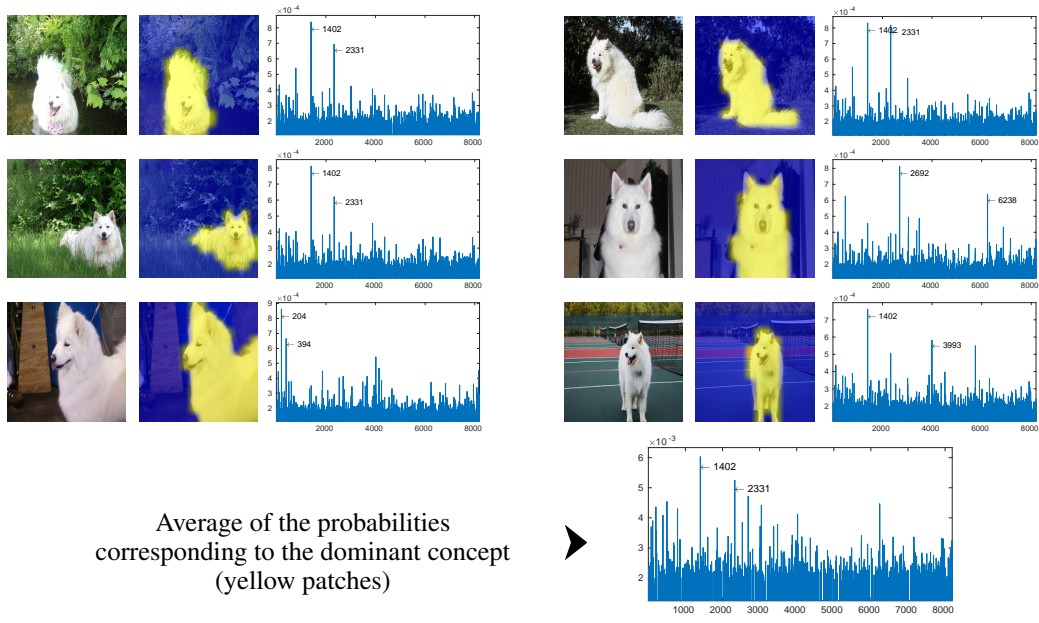

Figure 12: Concept representation across different images.

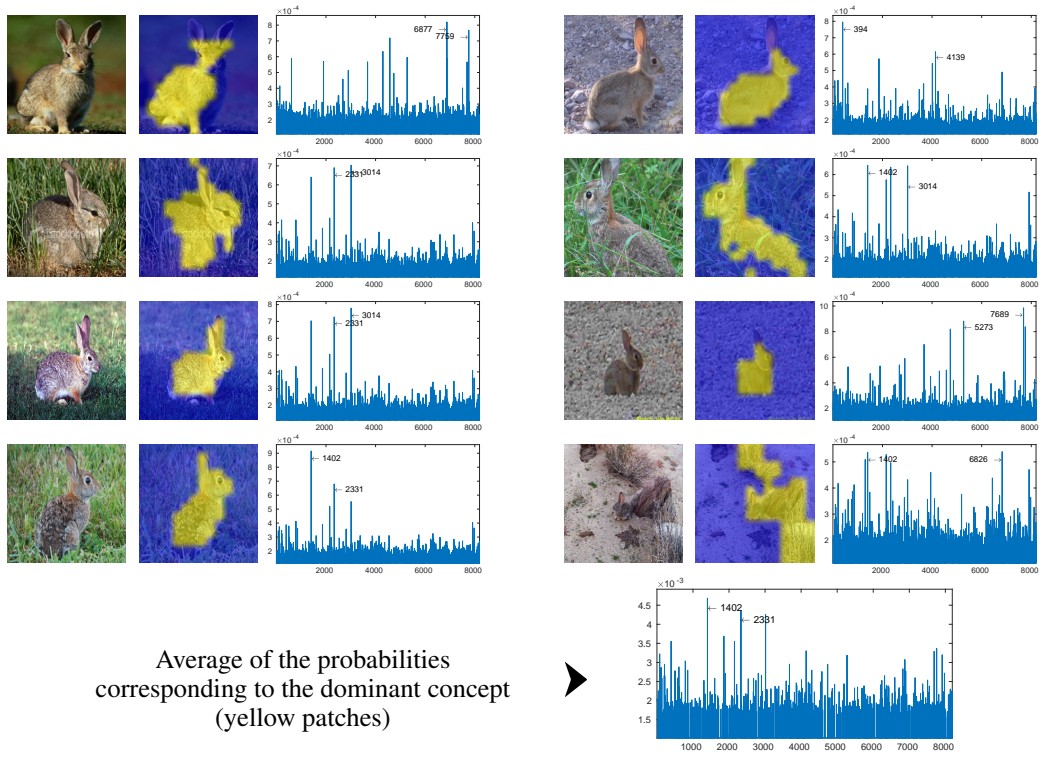

Figure 13: Concept representation across different images.

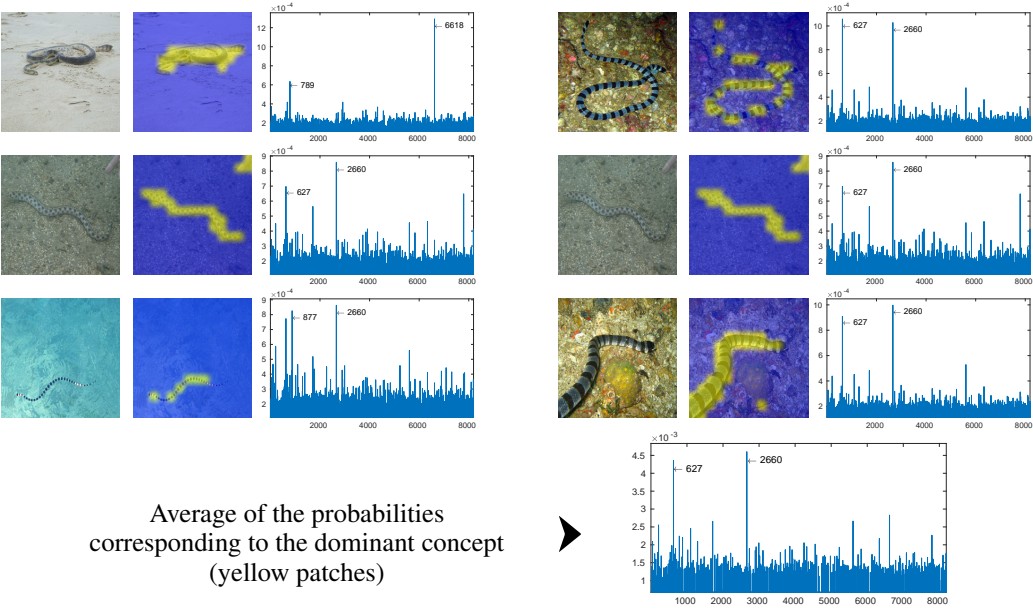

Average of the probabilities
corresponding to the dominant concept
(yellow patches)

Figure 14: Concept representation across different images.

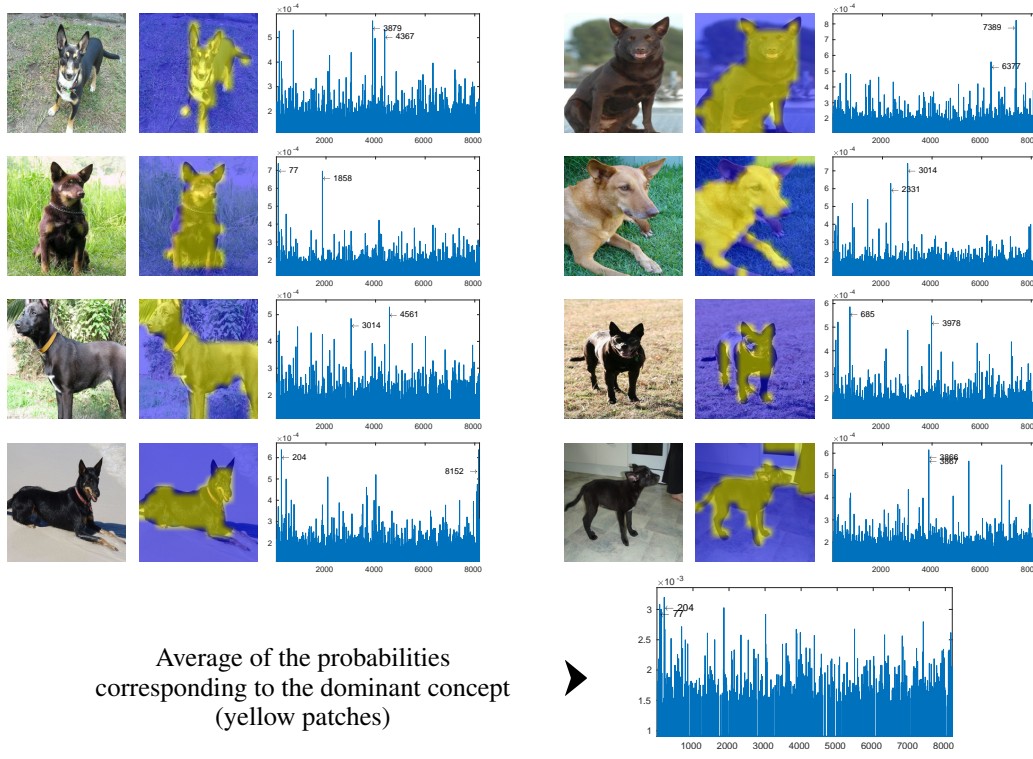

Average of the probabilities
corresponding to the dominant concept
(yellow patches)

Figure 15: Concept representation across different images.

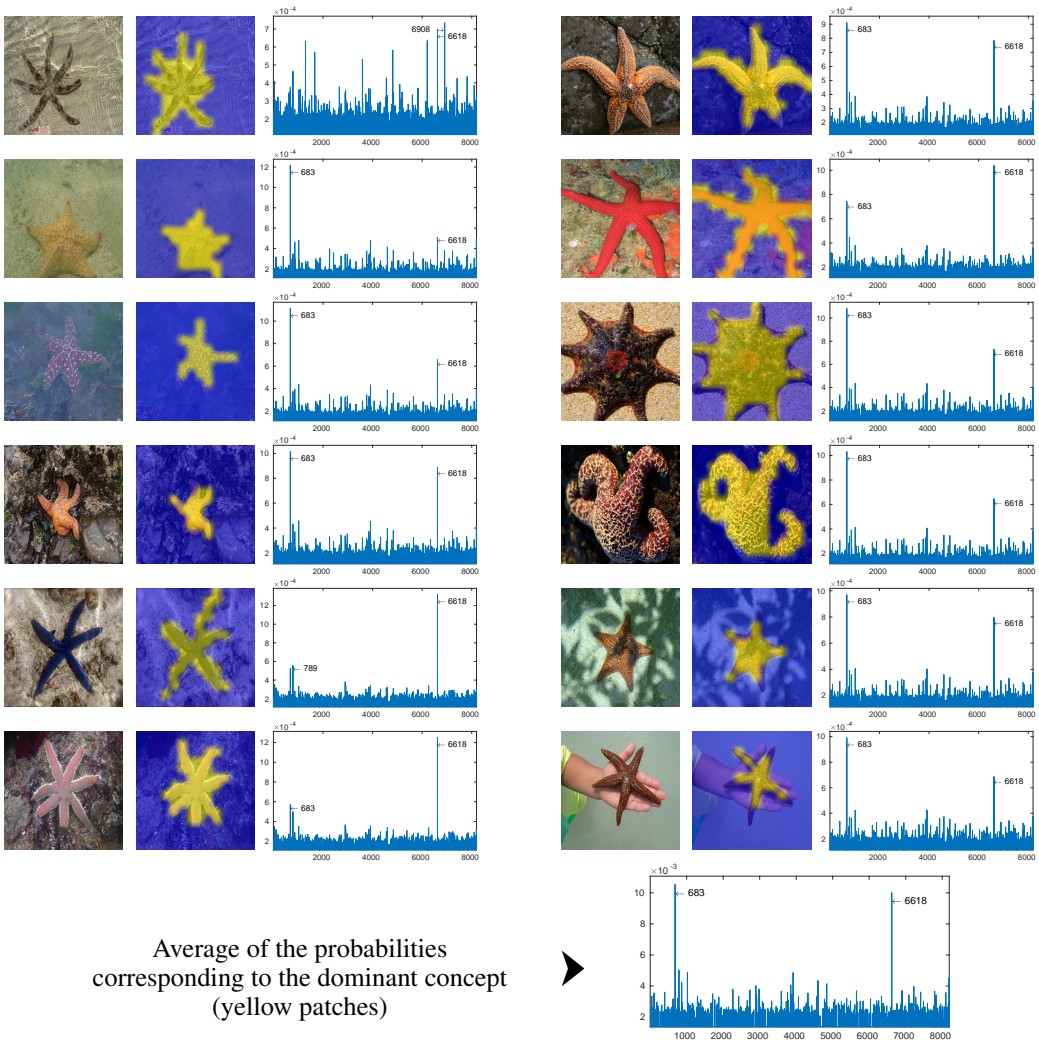

Average of the probabilities corresponding to the dominant concept (yellow patches)

Figure 16: Concept representation across different images.

