# OpenReview forum: "MC-SSL: Towards Multi-Concept Self-Supervised Learning"
_ICLR.cc/2023/Conference — Submitted to ICLR 2023_

### Official Review · Reviewer_rsxz · 2022-10-23

**Confidence:** 5
**Correctness:** 2
**Technical Novelty And Significance:** 2
**Empirical Novelty And Significance:** 2
**Recommendation:** 3

**Clarity, Quality, Novelty And Reproducibility:**

To me, the method is not novel enough and there are not a sufficient set of experiments to show the differences with the existing approaches.
 The implementation details are provided for the approach and it follows the same strategy as the existing approaches such as DINO, MAE, etc. I think that the method would be reproducible.

**Strength And Weaknesses:**

Strength:
	- A set of experiments have been conducted to demonstrate the effectiveness of the approach compared to some baselines.

Weaknesses:

1-  My main concern about this submission is the lack of novelty and proper comparison to SOTA. Recently, there are a bunch of well-known works existing in the literature with the same idea of self-supervised training of Transformers with masking and prediction. While I can see some discussions for BEiT and MAE, I can see only a footnote about iBOT as a very similar approach published in ICLR 2022 and almost no comparison with iBOT and BEiT in the experiments. These two approaches are very similar to the proposed one and there are only differences in defining tokenizer. I am not fully convinced by the note that the authors mentioned in the footnote about iBOT that it focuses on learning a dominant concept. I would like to see detailed discussions, experiments and proper comparisons to show the differences between the proposed and existing approaches.

2-  Comparisons on the object detection and semantic segmentation downstream tasks are not provided while the paper claims that the proposed approach would learn all semantic concepts in an image. This claim can be evaluated better on dense prediction tasks. Most of the existing approaches with the same idea such as MAE, iBOT, and BEiT have been evaluated on dense prediction tasks as downstream tasks.

3- I need more clarification on the "Patch Concept Learning" objective while we are feeding the image and its corrupted version to student and teacher networks. To me, it makes more sense if we learn the concept by feeding two different augmented views to the student and teacher and not the same view. This task to me seems like a trivial task at least for those patches which are not masked and are exactly the same for two networks. I would like the authors to clarify this in more detail.


**Summary Of The Paper:**

The paper presents an approach for self-supervised training of Transformers based on the idea of masking and predicting. Two objectives have been utilized during the training process. One is patch reconstruction and the other one is named "patch concept classification" to learn different concepts/classes for data tokens. The results of the paper on downstream tasks show some improvements compared to some of the existing approaches.

**Summary Of The Review:**

While there have been efforts to develop an idea for self-supervised pre-training of Transformers and evaluate it, however, there are some missing experiments that are very important to find out the benefit of this approach compared to the existing approaches in the literature. In my opinion, the paper needs major revision to fill those gaps I mentioned in the weaknesses.

---

> ### Author Response · Authors · 2022-11-18
> **Response to reviewer rsxz**
>
> We would like to thank the anonymous reviewer for the comments and suggestions. In the following, we tried our best to address the reviewer comment.
>
> * **The merits of MCSSL and how it is different from SiT/BEIT/MAE/SimMIM and iBot.**
>
> For the respect of the reviewer, we will add the BeiT results on ImageNet-1K which is 83.2% accuracy vs 84.0% accuracy of MCSSL when comparing ViT-B/16.
>
> According to footnote on page 6, we acknowledged that the MCSSL and iBot are parallel works (history can be checked easily that both works came to public domain in parallel and we can give the proof to SPC if needed). Nevertheless, we highlighted the difference between MCSSL and iBot. Both SiT and iBot have different focuses and intuitions, therefore, here we highlight the differences again between MCSSL and SiT and MCSSL and iBot.
>
> - GMML proposed in SiT lacks the ability to learn pseudo labels.
>
> - iBot extends the idea of GMML to incorporate DINO loss for recovery of missing information rather than reconstruction loss.
>
> - Both SiT and iBot use global embedding based contrastive heads which are SimCLR and DINO heads respectively. Due to which both SiT and iBot will focus on dominant objects mainly. We argue that this is one of the short coming of some of the existing self-supervised learning (SSL) methods which use global embedding based contrastive learning frameworks (more discussion is given below).
>
> - Unlike them MCSSL does not suffer from the problem of modelling mainly dominant objects. Moreover, MCSSL learns pseudo labels which will ensure that the learnt representation is consistent on multiple objects whether they are appearing within an image or happening across the images. Simple recovery of missing information from the context using GMML will not enforce this ability strongly. Please see updated Table 5 where we include the results for the MCSSL with global head on multilablel dataset PASCAL VOC. Note that the results drop by 1% by adding the global contrastive head.
>
> The patch concept learning loss can also be thought of online clustering of local dominant objects into self-learnt groups. Note that this does not affect the multi-concept modelling of MCSSL as one token consisting of 16x16 image patch that is most often will be predominantly from a single object. Since the local patch clustering is consistent across the images hence, we can say that there is semi-explicit mining of inter-image variance and invariance with online clustering. Please note that last year during the development of MCSSL we tried doing k-mean clustering explicitly in an end-to-end fashion. There was negligible difference in the performance, however, computational overhead was significantly more. Perhaps some kind of explicit mining of inter-image variance and in-variance can help after MCSSL pretraining for different tasks.
>
> * **Comparisons on the object detection and semantic segmentation downstream tasks**
>
> Thanks for the suggestion. We conducted some experiments on instance-segmentation on DAVIS 217 video object segmentation dataset as well as semantic segmentation on PASCAL VOC dataset and compared them to DINO, MoCo-v2, SwAV. For instance-segmentation we get comparable results to DINO as this task in doing instances segmentation of the dominant objects and it is expected that MCSSL would give comparable results. However, the real strength of MCSSL is not to capture one dominant concept but to capture multiple concepts. To verify this capability, we did semantic segmentation by freezing the ViT-S/16 backbone which is pretrained using MCSSL on ImageNet-1K and only trained a linear classifier, i.e., 1x1 Convolutional layer on top of the frozen data tokens. The mIOU results are given in the Table below. We can see the large margin in performance between state-of-the-art methods and MCSSL for semantic segmentation.
>
> | Method      | PASCAL VOC 2012 |
> |-------------|:---------------:|
> | MoCo-v2 [1] |       45.0      |
> | SwAV [2]    |       50.7      |
> | DINO [3]    |       50.6      |
> | MCSSL       |       66.4      |
>
> * **Objective of Patch Concept Learning**
>
> We agree with the reviewer that for visible patches the recovery of the information between teacher and student is trivial. In case of image reconstruction, we only reconstruct from the corrupted pixels. As for the patch classification, we agree that it will be better considering only the corrupted patches. Please note that in MCSSL framework, we are using GMML manipulation where 70% of the image is corrupted and the corruption is not necessary aligned with the token, which means that partial tokens can also be corrupted. To that end, most of the tokens in the student will be fully or partially corrupted. Thus, we believe that the effect of adding a constraint to only consider the fully/partially corrupted tokens to the loss will be negligible. Nevertheless, we agree with the reviewer that such constraint should be added.

---

### Official Review · Reviewer_85mW · 2022-10-24

**Confidence:** 5
**Correctness:** 2
**Technical Novelty And Significance:** 2
**Empirical Novelty And Significance:** 2
**Recommendation:** 5

**Clarity, Quality, Novelty And Reproducibility:**

There is self-contradiction in this paper draft. Check the weakness part for more details.

**Strength And Weaknesses:**

- Strength:
  - This paper organizes well with clear motivation and detailed experimental results.
  - The paper also explores the usage of popular self-supervised methods on data-scarce situations.

- Weakness:
  - Novelty is limited, especially compared with iBOT [1], which is published in ICLR 2022.
    - According to Table 5, the MC-SSL framework is exactly the same with iBOT combined with an extra auto-encoding reconstruction loss proposed by SiT.
    - According to the footnote in Page 6, the authors claim that the problem of iBOT is the usage of a global DINO head - "iBot focuses on learning a dominant concept via the classification token with DINO loss and hence, models the dominant class, which we think is a limitation of existing SSL methods.", which, however, is also used in MC-SSL according to Table 5. It seems that the multi-concept learning ability mainly comes from the usage of the reconstruction loss of SiT.
    - As iBOT is such an important baseline, it is not included in any of the experiments in Sec. 4.
    - The idea of combining reconstruction loss with patch-level contrastive learning has also been explored by some recent works [2,3].
  - Another important property of MC-SSL is the ability to learn multi-concept without annotations:
    - Different from [4, 5] which explicitly mine inter-image variance and invariance with K-means, MC-SSL only adopts two patch-level loss, which models inter-image variance and invariance in an implicit way (i.e., the intuition in Sec. 3.2). I wonder how an implicit way can achieve a better results than the explicit solutions in [4, 5]? Could you give more explanations about this problem?
    - It would be more interesting to report some numeric results to support the so-called multi-concept learning ability, like performance of unsupervised semantic segmentation.



[1] Zhou J, Wei C, Wang H, et al. ibot: Image bert pre-training with online tokenizer[J]. arXiv preprint arXiv:2111.07832, 2021.

[2] Dong X, Bao J, Zhang T, et al. Bootstrapped Masked Autoencoders for Vision BERT Pretraining[J]. arXiv preprint arXiv:2207.07116, 2022.

[3] Tao C, Zhu X, Huang G, et al. Siamese Image Modeling for Self-Supervised Vision Representation Learning[J]. arXiv preprint arXiv:2206.01204, 2022.

[4] Chen K, Hong L, Xu H, et al. Multisiam: Self-supervised multi-instance siamese representation learning for autonomous driving[C]//Proceedings of the IEEE/CVF International Conference on Computer Vision. 2021: 7546-7554.

[5] Hénaff O J, Koppula S, Shelhamer E, et al. Object discovery and representation networks[J]. arXiv preprint arXiv:2203.08777, 2022.



**Summary Of The Paper:**

This paper claims to achieve multi-concept learning without dependency on human annotation by applying two patch-level loss simultaneously, including the patch reconstruction loss from SiT and the self-distillation loss from DINO. Extensive experiments and visualizations demonstrate the effectiveness of the proposed method MC-SSL.

**Summary Of The Review:**

This is overall an interesting paper. However, the discussions can not be fully supported by the experiments. Furthermore, there exits self-contradiction and remaining unclear explanations in this paper draft version. I would like to see the author responses for more discussions.

---

> ### Author Response · Authors · 2022-11-18
> **Response to Reviewer 85mW**
>
> First, we would like to thank the reviewer for the thoughtful comments and the insightful reviews. We tried our best to address all the comments raised by the reviewer.
>
> * **Limited Novelty compared with IBot and SiT**
>
> According to page 6 footnote, we acknowledged that the MCSSL and iBot are parallel works (history can be checked easily to verify that both MCSSL and iBot came to public domain in parallel and we can give the proof to SPC if needed). Nevertheless, we highlighted the difference between MCSSL and iBot. Both SiT and iBot have different focuses and intuitions, therefore, here we highlight the differences again between MCSSL and SiT and MCSSL and iBot.
>
> - GMML proposed in SiT lacks the ability to learn pseudo labels.
>
> - iBot extends the idea of GMML to incorporate DINO loss for recovery of missing information rather than reconstruction loss.
>
> - Both SiT and iBot use global embedding based contrastive heads which are SimCLR and DINO heads respectively. Due to which both SiT and iBot will focus on dominant objects mainly. We argue that this is one of the short coming of some of the existing self-supervised learning (SSL) methods which use global embedding based contrastive learning frameworks (more discussion is given below).
>
> - Unlike them MCSSL does not suffer from the problem of modelling mainly dominant objects. Moreover, MCSSL learns pseudo labels which will ensure that the learnt representation is consistent on multiple objects whether they are appearing within an image or happening across the images. Simple recovery of missing information from the context using GMML will not enforce this ability strongly.
>
> Please note that one of the shortcomings of global representation based contrastive learning frameworks is that they model the dominant concept. The findings in the NIPS 2021 paper “Intriguing properties of contrastive losses.” [*] supports our claim and states in section 4.3, “The presence of dominant object suppresses the learning of features of smaller objects”. “However, for SimCLR, the learned representations of the smaller digit degenerate significantly when the size of the other digit increases, almost to the level of a random untrained network”. The clustering capability of contrastive approaches depends on the temperature parameter quite a lot. Lower temperature values can lead to over clustering and higher temperature values can lead to under clustering. Similarly, in Dino and iBot, based on the temperature hyperparameter of the teacher these global representations can describe either a single or very few dominant objects in the image. However, this problem is not highlighted in the current evaluation of SSL methods as all of them are evaluated on multi-class datasets including iBot.
>
> [*] Chen, Ting, Calvin Luo, and Lala Li. "Intriguing properties of contrastive losses." Advances in Neural Information Processing Systems 34 (2021): 11834-11845.
>
> The respectable reviewer may ask why we are submitting a year later, for the information we have submitted MCSSL last year but due to computational resources we were able to train ViT-S/16 for 400 epochs and ViT-B/16 for 300 epochs and did not show results using bigger models and bigger datasets which led to rejection of the paper last year. We hope we will not be penalized due to the lack of inability to train large models on large datasets.
>
> * **Usage of Dino Head in the architecture of MCSSL**
>
> The reviewer has made an important observation, we are stating that some of the global representation based contrastive frameworks model the dominant class or a couple of dominant classes. And yet we used global embedding (classification head) based contrastive loss. However, there is no contradiction as we used the global representation based contrastive head only for the cases of multi-class datasets where the focus of evaluation is on a single dominant object. We clearly stated that at the beginning of Section 4.1 “For the multi-class classification task, we include a class projection head similar to Dino in the pretraining stage to capture the general representation of the image”.
>
> The results in Table 5 are on multi-class datasets. But we agree that we should have added the results on multi-label datasets as well in this ablation. Due to the time constraints, we were only able to conduct experiments on Pascal dataset and we included it in Table 5. We will try to include VGenome and MSCOCO as well in the final version. Thanks again for the valuable suggestion.

---

> > ### Author Response · Authors · 2022-11-18
> > **Response to Reviewer 85mW - Cont.**
> >
> > * **Multi-concept ability mainly comes from SiT**
> >
> > Simple recovery by reconstruction error of missing information from the context using GMML will not enforce multi-concept learning ability particularly between different images. MCSSL learns a second task of token-wise pseudo labels which will ensure that the learnt representation is consistent on multiple objects whether they are appearing within an image or happening across the images. This is demonstrated with several visualization in appendices which shows consistent representations of data-token on objects whether they occur within an image or between different images. Please note that even with high variability in the objects we can get similar signatures (Figure 16 in appendix). Please note that all the visualizations in the papers and appendices are generated from ViT-S/16 without global representation head.
> >
> > * **combining reconstruction loss with patch-level contrastive learning**
> >
> > We agree with the reviewer that after MCSSL there are several recent works on arxiv which are trying to combine the idea of patch-level contrastive learning with reconstruction. We think this is one of the strengths of MCSSL as the idea is being validated by others “independently”.
> >
> > * **Explicit and implicit way of multi-concept learning**
> >
> > Thanks for initiating this interesting discussion. We agree that simple recovery of missing information from the context by reconstruction error using GMML will not explicitly ensure that the representations are consistent across the images (although it does that implicitly). However, the patch concept learning loss can also be thought of online clustering of local dominant object into self-learnt groups. Note that this does not affect the multi-concept modelling of MCSSL as one token consisting of 16x16 image patch most often will be predominantly from a single object. Since the local patch clustering is consistent across the images hence, we can say that there is semi-explicit mining of inter-image variance and invariance with online clustering. Please note that last year during the development of MCSSL we tried doing k-mean clustering explicitly in an end-to-end fashion. There was negligible difference in the performance, however, computational overhead was significantly more. Perhaps some kind of explicit mining of inter-image variance and invariance can help after MCSSL pretraining for different tasks.
> >
> > Nevertheless, while comparing the results qualitatively, between MCSSL and Multisiam (Figure 6 in Multisiam paper vs our visualizations) we see huge performance difference between k-mean clustering visualization, establishing the merits of MCSSL.
> >
> > * **performance of unsupervised semantic segmentation**
> >
> > Thanks for the suggestion. We conducted some experiments on instance-segmentation on DAVIS 217 video object segmentation dataset as well as semantic segmentation on PASCAL VOC dataset and compared them to DINO, MoCo-v2, SwAV. For instance-segmentation we get comparable results to DINO as this task in doing instances segmentation of the dominant objects and it is expected that MCSSL would give comparable results. However, the real strength of MCSSL is not to capture one dominant concept but to capture multiple concepts. To verify this capability, we did semantic segmentation by freezing the ViT-S/16 backbone  which is pretrained using MCSSL on ImageNet-1K and only trained a linear classifier, i.e., 1x1 Convolutional layer on top of the frozen data tokens. The mIOU results are given in the Table below. We can see the large margin in performance between state-of-the-art methods and MCSSL for semantic segmentation.
> >
> > | Method      | PASCAL VOC 2012 |
> > |-------------|:---------------:|
> > | MoCo-v2 [1] |       45.0      |
> > | SwAV [2]    |       50.7      |
> > | DINO [3]    |       50.6      |
> > | MCSSL       |       66.4      |
> >
> >
> > [1] Kaiming He, Haoqi Fan, Yuxin Wu, Saining Xie, and Ross  Girshick. Momentum contrast for unsupervised visual representation learning. In CVPR, 2020.
> >
> > [2] Mathilde Caron, Ishan Misra, Julien Mairal, Priya Goyal, Piotr Bojanowski, and Armand Joulin. Unsupervised learning of visual features by contrasting cluster assignments. In NeurIPS, 2020.
> >
> > [3]  Ziegler A, Asano YM. Self-Supervised Learning of Object Parts for Semantic Segmentation. In Proceedings of the IEEE/CVF CVPR 2022.

---

### Official Review · Reviewer_7m1Z · 2022-10-24

**Confidence:** 4
**Correctness:** 4
**Technical Novelty And Significance:** 3
**Empirical Novelty And Significance:** 4
**Recommendation:** 6

**Clarity, Quality, Novelty And Reproducibility:**

Clear, high quality paper. It's novel and useful for the community. Understanding and improving on the differences in what SSL models learn is important as SSL is becoming more and more ubiquitous. I see no code in supplementary material, which is a big drawback in such an empirical work.

**Strength And Weaknesses:**

Clear and nice paper. I think the presentation was sometimes slightly too high level. I'd like to see just a few more equations around the approach, and some clearer definitions.

Some other feedback:
* Figure 1 needs to be explained more, how many clusters were used etc, difference between second and third row.
* "they tend to encourage modelling of one dominant class per image using holistic representation and/or disregard the learning of contextual representations" where is the support of this claim?

**Summary Of The Paper:**

This paper introduces the MC-SSL method for self-supervised learning aimed at learning multiple concepts in images. It applies mainly two algorithmic techniques: Group Mask Model Learning (GMML) and learning of pseudo-concepts for data tokens using a momentum encoder framework. The paper is clear and interesting. The approach is novel and the results are good.

**Summary Of The Review:**

Strong paper with a novel technique (the aspect of pseudo-concepts on patch level) that's interesting and with great results. Given that this is an empirical paper, I believe code should be attached for reproducibility.

---

> ### Author Response · Authors · 2022-11-16
> **Response to Reviewer 7m1Z**
>
> First of all, thanks for the constructive feedback and the appreciation of the work. We attached the codes and will update the manuscript with more more equations to better describe the method mathematically.
>
> Regarding Figure 1, it is generated from ViT-S/16 pretrained without using any labels by using MCSSL. The model was pretrained using masked patch reconstruction and patch concept learning losses. Figure 1 shows the self-learnt grouping of representations learnt corresponding to data tokens by k-mean clustering. Figure 1 row 2 shows the clustering of representations corresponding to data tokens by asking for three clusters (assign k = 3 in k-means) and Figure 1 row 3 shows the clustering of data tokens by asking for four clusters. We did not use any additional processing of representation learnt from data token or showed the results by more advanced clustering algorithms as we just wanted to show the raw capability of the learnt representations on different objects, which appears to be consistent within an object. This is a strong point of the work over other self-supervised methods.  We will add details of network for Figure 1 to the manuscript.
>
>
> As for modelling the dominant concept, this shortcoming of contrastive learning is observed after mathematical analysis and our experiments in this paper support that. However, detailed mathematical discussion and experimental support for dominant object modeling by some of contrastive learning frameworks is out of scope for this paper and will be presented in a separate work. Very interestingly, after we conducted the MCSSL work we noticed that there is an empirical paper in NIPS 2021 which is highlighting this shortcoming. We have added the NIPS paper reference to the main manuscript. Please see section 4.3 in the NIPS 2021 paper “Intriguing properties of contrastive losses.” [*] which states, “The presence of dominant object suppresses the learning of features of smaller objects”.  “However, for SimCLR, the learned representations of the smaller digit degenerate significantly when the size of the other digit increases, almost to the level of a random untrained network. The clustering capability of contrastive approaches depends on the temperature parameter quite a lot. Lower temperature values can lead to over clustering and higher temperature values can lead to under clustering. DINO is a contrastive learning/clustering framework which tries to learn a single global representation/embedding of the whole image. Based on the temperature hyperparameter of the teacher these global representations can describe either single or very few dominant objects in the image.
>
>
> [*] Chen, Ting, Calvin Luo, and Lala Li. "Intriguing properties of contrastive losses." Advances in Neural Information Processing Systems 34 (2021): 11834-11845.

---

### Official Review · Reviewer_YV5g · 2022-10-30

**Confidence:** 3
**Correctness:** 3
**Technical Novelty And Significance:** 2
**Empirical Novelty And Significance:** 2
**Recommendation:** 5

**Clarity, Quality, Novelty And Reproducibility:**


This paper proposed an interesting SSL framework and presented some encouraging results regarding the learning of various concepts via their framework. However, there are still missing details regarding how the different methods are being evaluated, as well as some empirical evidence and discussion that their framework is truly capable of capturing concepts from images in various settings. In addition, the presentation and structures should be improved so that the readers can easily follow.

**Strength And Weaknesses:**

Strength:
- This work demonstrated its ability to differentiate concepts learned from images on small datasets while the majority of related works can only identify the dominant concepts from data.
- This framework avoids the common practice of using a large batch size for training in contrastive learning-based SSL frameworks, while still achieving relatively good performance.
- The authors conduct extensive experiments on multi-class and multi-label downstream tasks to demonstrate the effectiveness of the proposed framework in two settings.
- The results of visualized self-learned concepts are quite impressive and promising.

Weaknesses:
- The authors claim that their methods work well with small datasets. I am curious as to why not directly use smaller transformer models for these small datasets. How do the authors argue that these improvements indeed are from the proposed MC-SSL rather than related baseline approaches that are not well-trained?
- The notation of shared weights for prediction heads in Figure 2 is somewhat unclear. How exactly does each component share these weights?
- For the ablation studies, I am concerned about the 10% subset of ImageNet. The construction details of this dataset are missing. For example, a biased sampling strategy could cause the model to favor certain tasks or approaches downstream. This requires some clarification.
- How are the different approaches in Table 1 evaluated? Similarly, what trained models are used for Figure 1's visualization? This requires clarification. In addition, I wonder if we feed more data and train longer for the baselines, are they able to capture different concepts? Alternatively, the authors may want to show some learning curves to illustrate this.
- The presentation of this paper should be improved. For instance, the authors may want to highlight their most significant contributions or discuss the most significant differences between their works and those of SiT or DINO. Other minor issues: "our thesis is..." should be replaced with "our hypothesis is...". These notations should be consistent with "Dino → DINO".


**Summary Of The Paper:**

This paper introduces an SSL framework that is capable of learning different concepts in an image while the previous studies can only capture the dominant concepts. To this end, they proposed MC-SSL based on the components from SiT and DINO, i.e., Group Mask Model Learning (GMML) and Patch Concept Learning (PCL) based on knowledge distillation. In their evaluations, they demonstrated that MC-SSL can perform well with small datasets in multi-class and multi-label downstream tasks. Overall, this is an interesting work with some promising results.

**Summary Of The Review:**

Please check the strength and weaknesses for detailed comments.

---

> ### Author Response · Authors · 2022-11-16
> **Response to Reviewer YV5g**
>
> We would like to first thank the reviewer for the valuable feedback and the appreciation of the work. In the following, we address the raised concerns in detail:
>
> * **Smaller transformer models for small datasets? and How the approaches in Table 1 are evaluated?**
>
> In general, it is hard to train ViTs on small datasets due to the lack of inductive bias in transformers regardless of the model size. Several attempts have been made to tackle this problem [1,2,3]. Which is also proven from the poor performance when we them from scratch without any pre-training (Table1 – first row).
>
> In Table 1, we are using small variant of transformers, i.e. ViT-S/16, which has only 22M parameters. We could have also used ViT-T/16, the smallest variant of transformers with only 5M parameters, but to be fair with the SOTA where their parameters are adapted to bigger models, we decided to use ViT-S/16.
>
> As for the fair comparison with the SOTA, we pretrained all the methods for 3000 epochs except for MAE where we pretrained the model for 6000 epochs for fair comparison. We updated the manuscript to make it clear. Thanks.
>
> * **Related baseline approaches that are not well-trained?**
>
> For the comparison with SOTA, we employed the official codes with the suggested parameters by the authors and trained them on small datasets. Therefore, we argue that the improvements particularly for ViTs are indeed the result of better self-supervised pretraining which enables us to get SOTA performances on small datasets as well as on ImageNet-1K.
>
> * **Notation of shared weights**
>
> The shared weights are essentially 1x1 convolution layer. We chose to explain in terms of MLP due to the use of the term MLP in transformers. We note that all the MLP layers in transformers also share weights, so they are indeed 1x1 convolutional layers. After the introduction of transformers, the community is using the term ‘MLP’ more and more for 1x1 convolution in the context of transformers.
>
> * **Construction of the 10% of ImageNet**
>
> We chose 10% of the ImageNet samples randomly from each class, which we think would not introduce any bias toward a particular method. The purpose is to have a smaller but challenging dataset for ablations due to the limited GPU resources. We updated Section 4.3 of the main manuscript with the relevant discussion.
>
> * **Trained model for Figure 1's visualization?**
>
> Figure 1 is generated from ViT-S/16 pretrained without using any labels using MC-SSL. Note that we have two pretrained models, one for multi-class classification where class projection head, similar to DINO, is included in the pre-training stage as described in Section 4, and the other one is only using masked patch reconstruction and patch concept learning losses. For Figure 1, we employed the second pretrained model where the class projection head is not included. More details are added for Figure 1 to the manuscript to make it clear.
>
> * **Feeding more data and train longer for the baselines?**
>
> More data will help most of the self-supervised learning methods as well as the supervised methods. However, simply longer pretraining for self-supervised learning baselines may not help in capturing different concepts. For example, for visualization in Appendices, we use MC-SSL pretrained by us and for baseline we used official optimally pretrained models from the authors. We can see a clear advantage of MC-SSL in capturing different concepts qualitatively.
>
> For Table 2 and Table 3 we pre-trained MCSSL for 800 epochs for both ViT-S/16 and ViT-B/16. Other methods are also typically pretrained for 800 epochs except MAE which is pretrained for 1600 epochs and MoCo-v3 which is pretrained for 300 epochs. For MoCo-v3 pretraining for longer actually degrades the performance for downstream tasks, therefore, 300 epochs of pretraining are optimal. Hence, we think that simply longer pretraining may not help the baselines to capture different concepts.
>
>
> ###### [1] Li, Kehan, et al. "Locality guidance for improving vision transformers on tiny datasets." European Conference on Computer Vision. Springer (2022).
>
> ###### [2] Gani, Hanan, Muzammal Naseer, and Mohammad Yaqub. "How to Train Vision Transformer on Small-scale Datasets?." arXiv preprint arXiv:2210.07240 (2022).
>
> ###### [3] Lee, Seung Hoon, Seunghyun Lee, and Byung Cheol Song. "Vision transformer for small-size datasets." arXiv preprint arXiv:2112.13492 (2021).

---

> > ### Author Response · Authors · 2022-11-16
> > **Response to Reviewer YV5g - Cont.**
> >
> > * **Learning curves.**
> >
> > Most of the self-supervised losses saturate after a couple of hundred epochs. However, longer pretraining even after apparent saturation of self-supervised losses improves the results at the downstream tasks. Therefore, the meaningful learning curves would be to take self-supervised pretrained model after different epochs and finetune for downstream task and plot the performance to see how different models are converging and saturation of performance. We have shown it in ablation study for 10% of the ImageNet Figure 3 (b) of the manuscript. It can be seen clearly that longer pretraining would help MC-SSL more. But showing it on full ImageNet is prohibitive for us as that would involve finetuning multiple models on large scale datasets. We do not have access to intermediate models trained by other baselines works as the authors have provided their best pretrained models. Retraining these models for ImageNet to give meaningful learning curves is well out of our computational budget. For pretraining on ImageNet, we go with a standard number of pretraining which is used by other self-supervised pretraining methods, i.e., 800 epochs for pretraining.
> >
> > * **The significant difference between MC-SSL and those of SiT or DINO.**
> >
> > DINO is a contrastive learning/clustering framework which tries to learn a single global representation/embedding of the whole image. Based on the temperature hyperparameter of the teacher these global representations can describe either single or very few dominant objects in the image. SiT is the first working version of masked image modelling; however, it does not have notion of self learnt concepts. SiT does provide good pretraining tasks but will not be directly applicable to capturing different concepts and particularly assign a common pseudo label to the captured concepts across different images. MCSSL recovers the missing information but in addition to that captures different concepts within the image. More importantly, different instances of the same object have consistent representations if they appear within an image or across different images. We will update the manuscript accordingly to state the significance of our proposed approach.
> >
> > * **Minors**
> >
> > We have changed the manuscript based on the suggestions from the reviewer.  Thanks a lot.

---

### Decision · Program_Chairs · 2023-01-20

**Decision:**

Reject

**Justification For Why Not Higher Score:**

This paper presents several issues - (1) lack of proper empirical comparisons; (2) ablation analysis; (3) correctly evaluating the central claim in the paper. Reviewers are aligned in not accepting the paper.

**Justification For Why Not Lower Score:**

N/A

**Metareview: Summary, Strengths And Weaknesses:**

*Summary*: This paper proposes to use a mixture of recontruction and self-distillation losses to learn an image representation. The authors argue that general self-distillation losses use a global feature and thus focus only on single dominant objects. In contrast, the authors propose - (1) patch-level self-distillation loss; (2) l1 reconstruction loss. The method is evaluated on image classification benchmarks - multiclass and multilabel settings. In the comments, the authors also provide segmentation results by training a linear classifier.

*Strengths*: (1) The problem setting studied in this paper is quite important. Global level losses indeed focus on single objects, which limits their applications. (2) Extensive empirical evaluation on multiple different datasets and settings. (3) Ablation experiments bring out the importance of image corruption, epochs, output dimension, and global feature learning.

*Weaknesses*: (1) The central claim of the paper is that the proposed loss function enables multi-concept learning. The reviewers (rsxz, 85mW) , and the AC feel that this claim is under-evaluated. Datasets used in Table 1,2,3 (and majority of 5) are about single object recognition. The only results relevant to the claim are in Table 4. This table does not compare against other methods that use a patch-based loss (MAE, iBOT, data2vec, SimMIM, BeIT). (2) As pointed out by reviewer 85mW, it is unclear what enables MC-SSL's multiconcept learning - is it the reconstruction loss; the patch self-distillation loss; both?. Table 5 lacks this analysis. The relevant analysis would disable the reconstruction loss from MC-SSL, and the resulting method would be similar to iBOT. (3) The model settings used vary according to the dataset  and Table 5 shows that the global head actually hurts multi-label performance on VOC. This implies that the proposed solution needs to be catered heavily to the downstream task, a property not noted in the work.